# Optogenetic dissection of Rac1 and Cdc42 gradient shaping

S. de Beco[1], K. Vaidžiulytė[1], J. Manzi[1], F. Dalier[2], F. di Federico[1], G. Cornilleau[1], M. Dahan[1] & M. Coppey[1]

During cell migration, Rho GTPases spontaneously form spatial gradients that define the front and back of cells. At the front, active Cdc42 forms a steep gradient whereas active Rac1 forms a more extended pattern peaking a few microns away. What are the mechanisms shaping these gradients, and what is the functional role of the shape of these gradients? Here we report, using a combination of optogenetics and micropatterning, that Cdc42 and Rac1 gradients are set by spatial patterns of activators and deactivators and not directly by transport mechanisms. Cdc42 simply follows the distribution of Guanine nucleotide Exchange Factors, whereas Rac1 shaping requires the activity of a GTPase-Activating Protein, β2-chimaerin, which is sharply localized at the tip of the cell through feedbacks from Cdc42 and Rac1. Functionally, the spatial extent of Rho GTPases gradients governs cell migration, a sharp Cdc42 gradient maximizes directionality while an extended Rac1 gradient controls the speed.

---

[1] Laboratoire Physico Chimie Curie, Institut Curie, PSL Research University, Sorbonne Université, CNRS, 75005 Paris, France. [2] PASTEUR, Département de chimie, École normale supérieure, CNRS UMR 8640, PSL Research University, Sorbonne Université, 75005 Paris, France. Correspondence and requests for materials should be addressed to M.C. (email: mathieu.coppey@curie.fr)

Cell migration plays a major role in various biological functions, including embryonic development, immune response, wound closure, and cancer invasion. Cells, either isolated or in cohesive groups, are able to respond to many types of spatially distributed environmental cues, including gradients of chemoattractants[1,2], of tissue stiffness (durotaxis)[3–5], and of adhesion (haptotaxis)[6,7]. To sense and orient their migration accordingly, cells need to integrate complex and noisy signals and to polarize along the selected direction. A simple explanation for such directed migration would be to consider that external gradients are directly translated into internal gradients. However, recent works[8–10] point to a two-tiered mechanism. First, a set of signaling proteins (Rho GTPases and Ras) behave as an excitable system that spontaneously establish intracellular membrane-bound gradients, conferring the ability of cells to polarize even in the absence of external stimuli. Second, a sensing machinery based on membrane receptors aligns the polarization axis along the direction of external gradient cues. In the present work, we address the mechanisms shaping the Rho GTPases gradients at the front of randomly migrating cells.

Rho GTPases are known to play a key role in orchestrating the spatially segregated activities that define the polarity axis of migrating cells. At the cell front, membrane protrusions fueled by actin polymerization push the cell forward, while retraction of the cell back depends on acto-myosin contractility[11–13]. The schematic view is that front-to-back gradients of Cdc42 and Rac1 define the cellular front, while RhoA is mostly active at the back. Cdc42 is known to be required for filopodia formation, through N-WASP-mediated activation of the ARP2/3 complex as well as F-actin bundling proteins such as fascin and formin[11,14]. Conversely, Rac1 is involved in branched actin polymerization and lamellipodia formation, through WAVE-mediated activation of the ARP2/3 complex[15]. RhoA is responsible for stress fiber formation and retraction of the cellular tail through Rho kinase-mediated contraction of myosin II[16,17]. In reality the situation is more complex since RhoA is also active at the very front of migrating mouse embryonic fibroblasts[18,19] and is involved in actin polymerization through Diaphanous-related formins as well as focal adhesions[20,21]. In addition, the Rho GTPase family contains more than the three members aforementioned, with more than 20 proteins having been discovered[20,22]. Despite the fact that these other members are classified in the three Cdc42, Rac1, and RhoA sub-families, they present overlapping activities.

Three main classes of proteins regulate the activity of Rho GTPases. Guanine Exchange Factors (GEFs) activate Rho GTPases by promoting the exchange from GDP to GTP, whereas GTPase-activating proteins (GAPs) inhibit Rho GTPases by catalyzing the hydrolysis of GTP[23]. A multitude of GEFs and GAPs ensure signaling specificity and fine-tuned regulation. In addition, guanine-nucleotide dissociation inhibitors (GDIs) are negative regulators of Rho GTPases, extracting them from the plasma membrane and blocking their interactions with GEFs[24,25]. GEFs and GAPs can be localized and activated by upstream factors such as receptor tyrosine kinases or interaction with lipids such as PIP3[26,27], hereby connecting the polarization machinery with the sensing one. Moreover, complex crosstalks connect Rho GTPases and their interactors, resulting in a signaling network that finely regulates Rho GTPases activities. Although many molecular interactions defining this signaling network have been characterized, we currently have little insight on how these interactions are orchestrated in space to shape Rho GTPase activity patterns.

Positive feedbacks acting on Rac1, Cdc42, and RhoA have been proposed to account for their ability to form gradients spontaneously. Rho GTPase activity pulses would be generated thanks to an excitable system[9] and specific activators like GEFs would orient and stabilize them[8,28]. Yet, activity patterns governed by excitable systems have a propensity to propagate through the whole cell, and inhibitory mechanisms are required to limit their expansion[29–31]. Three mechanisms could confine Rho GTPases activities. First, Rho GTPase cycles can be locally regulated by GEF and GAP concentrations, whose distributions along the cell would shape Rho GTPase intracellular gradients[31–34]. Second, anchoring or trapping in the cortical acto-myosin network can decrease diffusion considerably. Since Rho GTPases trigger actin polymerization and branching, this mechanism could act as a negative feedback restricting their activity zones. Third, Rho GTPase extraction from the plasma membrane by GDIs can be locally regulated[25], such that deactivation regions could be set by the activity of GDIs. It is unclear which of these mechanisms is responsible for the formation of Rho GTPase intracellular spatial patterns.

In this work, we show that Cdc42 and Rac1 gradients are formed thanks to a combination of distributed GEFs and GAPs and not directly by diffusion or actin retrograde flow from a localized source. A combination of experimental approaches and minimal mathematical model suggests that: (i) the amount of active Cdc42 simply follows its GEFs distribution thanks to a uniform GAP activity, (ii) the Rac1 gradient requires an additional inhibition at the front by the β2-chimaerin GAP that shifts its peak of activity and hereby increases its spatial extent. We show that the localized activity of β2-chimaerin depends on both Cdc42 and Rac1, forming a negative feedback on Rac1, and that the actin retrograde flow is required for β2-chimaerin enrichment. Finally, we show that the resulting spatial properties of Cdc42 and Rac1 gradients govern the directionality and the speed of cell movement, respectively.

## Results

**Cdc42 and Rac1 gradients show two distinct shapes at the front of migrating cells.** We investigated the spatial activity gradients of Cdc42 and Rac1 Rho GTPases at the basal plasma membrane by imaging FRET biosensors based on an intramolecular fusion between Rac1 and a PAK1 binding domain[35]. HeLa cells stably expressing FRET reporters were left to migrate randomly on glass coverslips, and were imaged using total internal reflection fluorescence (TIRF) microscopy. The FRET ratio was calculated as a proxy for GTPase activity. Front-to-back gradients of either Cdc42 or Rac1 activity were measured from the cell protruding edge to the nucleus (Fig. 1a). As previously reported in neutrophils[9], we observed gradients that differed both in shape and in spatial extent. Cdc42 gradient was steep and monotonous, peaking at the protruding edge, and presenting an exponentially decaying profile of characteristic length $d = 8.3\,\mu m \pm 0.6\,\mu m$ (SEM, $n = 19$). In contrast, Rac1 gradient peaked at a distance $d = 5.8 \pm 0.5\,\mu m$ from the cell edge, and then decayed with a characteristic length $d = 9.6\,\mu m \pm 0.7\,\mu m$ (characteristic length of the exponentially decaying part, $n = 31$). We defined the extent of the gradient by the distance between the tip of the cell and the point where the signal reaches half-amplitude. The extent for Rac1 was $d = 14.6 \pm 0.7\,\mu m$, compared to $d = 8.9 \pm 0.6\,\mu m$ for Cdc42 (Fig. 1b). Interestingly, these observations match those reported for gradients in other cell lines[9,36]. We thus questioned what could be the mechanisms generating these gradients and accounting for their distinct shapes.

Two generic classes of models can account for the patterning of spatially graded distributions[29]. The first class relies on transport mechanisms (diffusion, flow) to establish gradients from a localized source (Fig. 1c, d). A canonical example is the synthesis−diffusion−degradation model, which has been heavily discussed in the context of the Bicoid morphogen gradient[37]. The

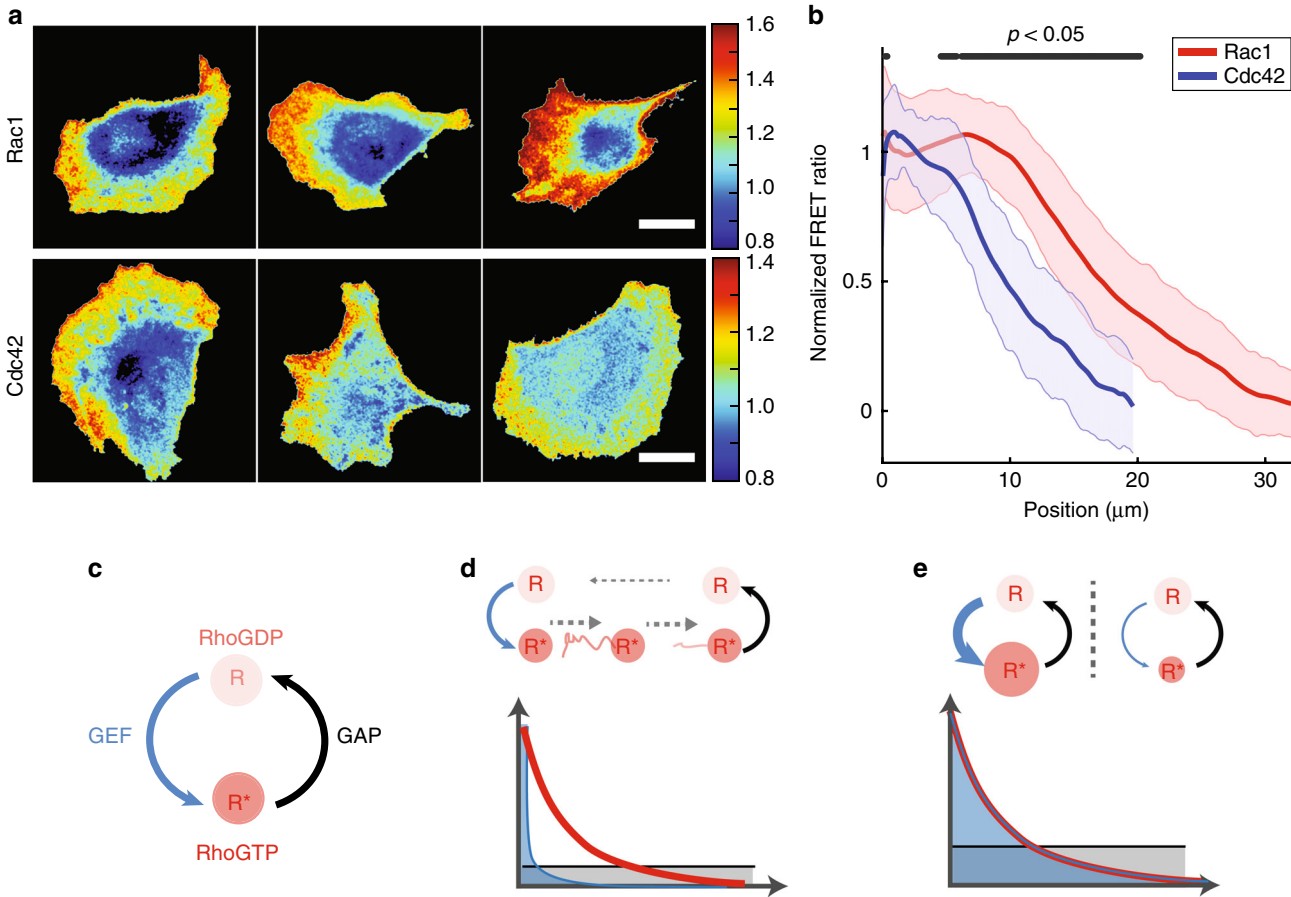

**Fig. 1** Rac1 and Cdc42 activity gradients have different shapes. **a** FRET biosensors were used to monitor Rac1 (top) or Cdc42 (bottom) activity in freely migrating HeLa cells. GTPase activity is measured by the FRET ratio, and represented with a color scale. Several representative cells are shown. Scale bar: 20 µm. **b** Mean normalized FRET ratio of Rac1 (red) and Cdc42 (blue) is plotted as a function of the distance from the cell edge. The error bars indicate the standard deviation (s.d.) of $n = 31$ (Rac1) or $n=19$ (Cdc42) cells. Black segments at the top show positions at which the curves are statistically different ($p < 0.05$, Wilcoxon's rank sum test). **c** Rho GTPase cycle, where the protein switches between an inactive and active state thanks to activators (GEFs) and deactivators (GAPs). **d**, **e** Two simplified mechanisms can explain the formation of cellular-scale Rho GTPase gradients. **d** A sharply localized GEF (blue profile) acts as a punctual source of active Rho GTPases (red) that are further transported by diffusion or flow (dashed gray arrows) until they reverse to the inactive state thanks to a GAP (black). **e** A cellular-scale distributed GEF locally activates the Rho GTPase such that both have the same profile

second class of models assumes a graded distribution of activators and deactivators (Fig. 1c, e). In this context, the local concentration is set by the local balance between activation and deactivation. This second class of model has also been proposed to explain the establishment of morphogen gradients, e.g. for the formation of the bone morphogenetic protein gradient that patterns the dorso-ventral axis of the early *Xenopus* embryo[38,39].

**Cdc42 and Rac1 gradients are shaped by spatially distributed GEFs and GAPs but not by diffusion**. In order to distinguish between these two classes of models, we opted for an input −output relationship approach. We used optogenetics[40,41] to impose activation gradients of either Intersectin-1 (ITSN) or T-Cell Lymphoma Invasion and Metastasis 1 (TIAM1), two GEFs specifically activating Cdc42 or Rac1, respectively. We used fusions of CRY2 with the DHPH catalytic domain of ITSN or TIAM to activate specifically Cdc42 or Rac1[41] (Fig. 2c). A home-made illumination setup using a DMD (Digital Micromirror Device[42]) allowed us to shine spatial gradients of light with an 8-bit gray level resolution. Cells were confined on round micro-patterns to prevent cell shape polarity[43] and gradients of light with slopes ranging from 1× to 4× were applied (Fig. 2a). As we could predict in a previous work[44], recruitment of the

cytoplasmic optogenetic partner CRY2 to the basal plasma membrane followed the stimulation signal with the addition of an exponential decaying tail of 5 µm characteristic length due to the diffusion of CIBN-CRY2 dimers at the membrane (Fig. 2b). This allowed us to tune precisely the spatial distribution of desired GEFs and test the relationship between the activation input and the output in terms of GTPase activity. If any transport mechanism (model 1) was taking place, we would expect a difference in the spatial distribution of the output compared to the input. For example, diffusion would give rise to a more extended distribution of the output by the addition of a length scale $\ell_{\text{diff}} = \sqrt{\tau D}$, where $\tau$ is the lifetime of the Rho GTPase in its active GTP-bound state and $D$ is its lateral diffusion coefficient. Contrarily, model 2 predicts that the spatial distribution of the output would mirror the distribution of the input, given that deactivators are uniform. Indeed, we reasoned that the optogenetic activation would dominate the other sources of activation such that the input−output relationship would reveal the distribution of the deactivators.

To determine whether Cdc42 and Rac1 followed the imposed activation pattern, we used downstream effectors as reporters of GTPase activity. The protein PAK1 is activated downstream of both Cdc42 and Rac1. We monitored the basal membrane recruitment of a PAK1-iRFP fluorescent reporter following

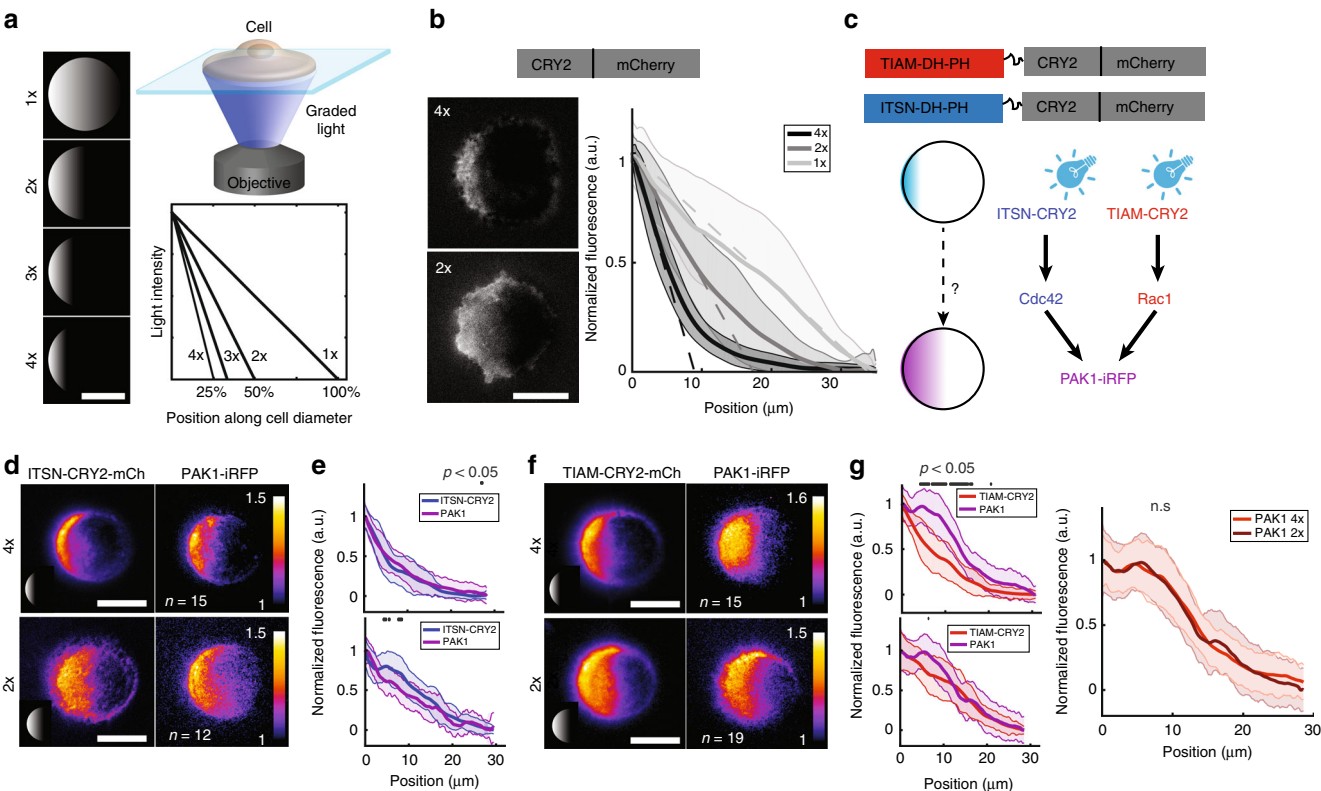

**Fig. 2** Cdc42 and Rac1 have different responses to GEF activation. **a** We imposed GEF activity gradients of different slopes using optogenetics. Patterned illumination with grayscale levels (left) was shone onto the samples, imposing linear light gradients of the same amplitude but different spatial extents on cells attached on round micropatterns of 35 μm in diameter (top right). The reference gradient called 1× spans over the whole diameter of the cell. The gradient 2× spans over a half cell diameter, and therefore has a slope twice as sharp as for the gradient 1× (bottom right). **b** Membrane recruitment of the optogenetic partner CRY2-mCherry to the basal side of cells on round micro-patterns following 30 min of illumination with 4× (top left) and 2× (bottom left) gradients. mCherry fluorescence (solid lines) was measured along the cell diameter following illumination with 1× (light gray), 2× (medium gray) and 4× (black) gradients (dashed lines). Error bars indicate the s.d. of $n = 15$ (1×), $n = 24$ (2×) and $n = 15$ (4×) cells. **c** We activated GEFs of Cdc42 (ITSN) or Rac1 (TIAM) with light gradients and measured the fluorescence pattern of PAK1-iRFP. Schemes at the top represent the fusion proteins used. **d** PAK1-iRFP recruitment following 4x (top) or 2× (bottom) activation gradients of ITSN. Fluorescence was recorded using TIRFM in HeLa cells on round micro-patterns, and initial fluorescence was subtracted for normalization. Micrographs represent the averaged fluorescence of $n = 15$ (4×, top) or $n = 12$ (2×, bottom) cells. Insets show the illumination patterns (not to scale). **e** Normalized fluorescence of ITSN-CRY2-mCherry (blue) and PAK1-iRFP (purple) was measured along the cell diameter and averaged (solid lines). Error bars: s.d. Gray lines at the top show positions at which the curves are statistically different ($p < 0.05$, Wilcoxon's rank sum test). **f** PAK1-iRFP recruitment following 4× (top, $n = 15$) or 2× (bottom, $n = 19$) activation gradients of TIAM. **g** Normalized fluorescence of TIAM-CRY2-mCherry (red) and PAK1-iRFP (purple) along the cell diameter of $n = 15$ (4×, top) or $n = 19$ (2×, bottom) cells, Error bars: s.d. Gray: Wilcoxon's rank sum test ($p < 0.05$). Single cell data for **e** and **g** are presented in Supplementary Figure 3. Scale bars: 20 μm

gradient activation of each one of the GTPases (Fig. 2c). As controls, we verified that the observed recruitment of PAK1-iRFP was not due to fluorescence bleed-through or nonspecific activity of CRY2-mCherry (Supplementary Figure 1a), nor to volume effects or cell deformation (Supplementary Figure 1b). Importantly, we also verified that the GEF DHPH domains used in our optogenetic approach were truly specific (Supplementary Figure 2). PAK1-iRFP recruitment patterns followed the activation gradients of ITSN-CRY2 remarkably well, independently of their spatial extents (Fig. 2d, e, Supplementary Movie 1). We could not detect any significant difference between the PAK1 response and the activating ITSN gradients, independently of their spatial extents (Fig. 2d, e, Supplementary Movie 1), up to the resolution of our measurement estimated as ~2 μm (two standard deviations of the spatial extent). This result suggests that GEF activity levels are sufficient to shape Cdc42 activity patterns without the requirement of other mechanisms. Conversely, PAK1-iRFP spatial recruitment was independent of the shape of the activating TIAM-CRY2 gradient. It did not follow the sharpest activation gradient (4×), and the peak at 6 μm from the protrusion edge was

present from the beginning of the stimulation (Supplementary Figure 4, Supplementary Movie 2) despite its absence from the gradients of TIAM-CRY2 (Fig. 2f, g). Interestingly, the PAK1-iRFP gradient obtained with our synthetic approach matched the Rac1 gradient observed in native cells (Fig. 1b). We thus sought to discriminate between two possibilities explaining how the Rac1 gradient is shaped: whether shaping involves transport or nonuniformly distributed deactivators.

**A crosstalk between Cdc42 and Rac1 through GEFs and GAPs contributes to Rac1 gradient shaping.** A complex crosstalk between the Cdc42 and Rac1 pathways has been shown previously[12,45]. We questioned whether such network could explain the complex pattern of Rac1 activity we observed. We used the Abi1-iRFP fusion protein as a reporter of Rac1 activity. Abi1 is part of the WAVE complex that has been shown to be activated specifically by Rac1 but not by Cdc42[46] (Fig. 3a). We observed that Abi1 is activated at the cell edge following TIAM but also ITSN optogenetic activation (Fig. 3b), suggesting that Cdc42 directly or indirectly activates Rac1. Interestingly, in both

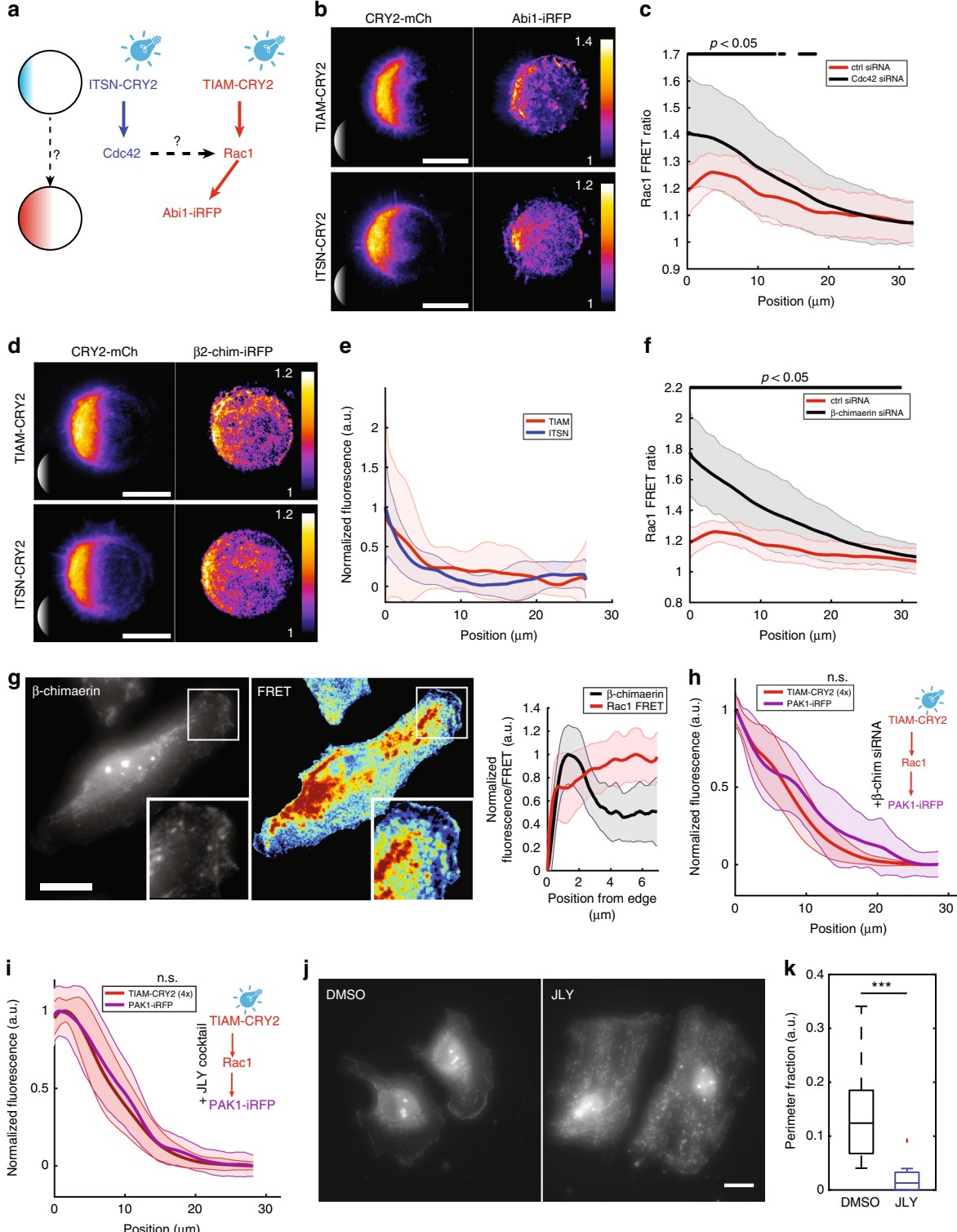

cases the induced Abi1 recruitment was more restricted to the cell border than the activating gradients (Supplementary Figure 5). This observation is in accordance with the known distribution of the WAVE complex at the tip of the lamellipodia[47], suggesting a compartmentalization independent of the immediate Rho GTPase activation. Yet, in addition to the positive crosstalk, we also observed a cross inactivation of Rac1 by Cdc42. When we inhibited Cdc42 by siRNA (Supplementary Figure 6a), we observed an increase of Rac1 activity at the cell front as measured by the non-normalized FRET profile (Fig. 3c). Strikingly, the

bump of Rac1 activity 6 μm from the cell edge was abolished in this condition. Since the overall effect of Cdc42 depletion is to increase Rac1 activity, we reasoned that the dominant role of Cdc42 on Rac1 is to specifically activate a GAP inhibiting Rac1.

β2-chimaerin is a GAP of Rac1 that was shown to be activated at the protrusion edge downstream of chemotactic signals[48]. We monitored the recruitment of the β2-chimaerin-iRFP reporter following the optogenetic activation of TIAM or ITSN. We could observe that both pathways could recruit β2-chimaerin at the cell edge, in a very localized fashion similar to the WAVE recruitment

**Fig. 3** Cdc42 and β2-chimaerin are involved in shaping the activity gradient of Rac1. **a** We activated GEFs of Cdc42 (ITSN) or Rac1 (TIAM) with light gradients and measured the fluorescence pattern of Abi1-iRFP. **b** Averaged Abi1-iRFP recruitment (right column) following 4× activation gradients (left column) of TIAM ($n = 10$, top) or ITSN ($n = 11$, bottom) visualized using TIRFM on round micro-patterns. The averaging procedure is explained in the Methods section. Insets show the illumination patterns (not to scale). **c** Non-normalized Rac1 FRET ratio profiles along cell diameters of cells treated with control siRNA ($n = 38$, red) or Cdc42-directed siRNA ($n = 37$, black). Error bars: s.d. Gray lines at the top show positions at which the curves are statistically different ($p < 0.05$, Wilcoxon's rank sum test). **d, e** β2-chimaerin-iRFP (right) recruitment following 4× activation gradients (left) of TIAM ($n = 11$, top) or ITSN ($n = 12$, bottom), imaged in TIRFM on round micro-patterns. **d** Micrographs represent the averaged fluorescence (see Methods). **e** Normalized fluorescence of β2-chimaerin-iRFP was measured along the cell diameter following the activation of ITSN-CRY2-mCherry (blue, $n = 12$) or TIAM-CRY2-mCherry (red, $n = 11$). Error bars: s.d. **f** Non-normalized Rac1 FRET ratio profiles along cell diameters of cells treated with control siRNA ($n = 38$, red) or β2-chimaerin-directed siRNA ($n = 25$, black). Error bars: s.d. Gray: Wilcoxon rank sum test ($p < 0.05$). **g** β2-chimaerin staining (left, Gamma correction was applied to images in order to visualize the full dynamics) compared to normalized Rac1 FRET (middle) in the same cells. Insets show zoomed regions of the cell edge. Levels of β2-chimaerin (black) and Rac1 activity (red) are anticorrelated at the cell front (right panel, $n = 8$, Error bars: s.d.). **h** PAK1-iRFP (purple) recruitment following 4× activation gradients of TIAM (red) after treatment with β2-chimaerin-directed siRNA ($n = 14$). Curves were found not significantly different on their whole length (Wilcoxon, $p > 0.05$). **i** PAK1-iRFP (purple) recruitment following 4× activation gradients of TIAM (red) after treatment with JLY cocktail ($n = 23$). n.s., nonsignificant. **j** β2-chimaerin staining after DMSO (left) or JLY cocktail (right) treatment. **k** Fraction of cell perimeter showing β2-chimaerin signal at the cell edge larger than in the cytosol (DMSO: $n = 11$, JLY: $n = 10$). Fluorescence at the cell edge was measured along a 1-μm-thick line obtained from the thresholding-based segmentation of the cell shape. The signal in the cytosol was evaluated from a 1 μm-thick line outlining that cell edge on its cytosolic side. Box plots represent the median, interquartile (box), 1.5 IQR (whiskers), and outliers (red crosses). Statistical significance was evaluated using Wilcoxon's rank sum test. ***$p \leq 0.001$. Scale bars: 20 μm

(Fig 3d, e), suggesting that this GAP recruitment is conditioned by other signaling components belonging to the tip of the lamellipodia, in accordance with the colocalization between β2-chimaerin and F-actin observed in the lamellipodia of unstimulated cells (Supplementary Figure 7). Accordingly, inhibiting β2-chimaerin using siRNA (Supplementary Figure 6b) led to a strong increase of Rac1 activity especially at the cell front such that the bump was abolished (Fig. 3f). This result suggests that β2-chimaerin might act downstream of Cdc42 and Rac1 to inhibit Rac1 locally at the cell front. Indeed, at the front of randomly migrating cells we observed an anticorrelation between β2-chimaerin and Rac1 activities measured by FRET (Fig. 3g). We could verify that the observed localization of β2-chimaerin at the cell edge was not due to volume effects related to the local membrane ruffling activity (Supplementary Figure 8). We further confirmed the direct role of β2-chimaerin in shaping the Rac1 gradient by inducing the sharp Rac1 activation (4×) using optogenetics in β2-chimaerin-depleted cells, which resulted in a PAK1 gradient that now matched the activating profile (Fig. 3h).

Orthogonally to the previous experiments, we also tested the role of transport in shaping the Rac1 gradient. Given that we observed the same PAK1 spatial profile for two distinct TIAM-CRY2 activating gradients (Fig. 2g), we excluded diffusion as it would have smoothened both input distributions. Conversely, the retrograde flow of actin in the lamellipodia can give rise to two similar outputs if the distribution of the flow velocities is ultimately limiting the spatial expansion of the gradient. When cells were treated with the Jasplakinolide-LatrunculinB-Y27632 (JLY) drug cocktail that freezes actin dynamics[49], we indeed observed a Rac1 activity gradient that matched the sharp (4×) TIAM-CRY2 input gradient (Fig. 3i). Surprisingly, this result shows that the actin retrograde flow can also account for the bump observed in the endogenous Rac1 gradient besides our previously found role for β2-chimaerin. However, this effect could be indirect if the retrograde flow was acting not on Rac1 itself but on the machinery required for proper β2-chimaerin localized distribution. To test this hypothesis, we compared the distribution of β2-chimaerin in control and JLY-treated cells (Fig 3j). β2-chimaerin localization disappeared from the tip of migrating cells in JLY-treated cells, confirming the indirect role of actin dynamics (Fig. 3k).

**A minimal model of local reactions recapitulates Cdc42 and Rac1 gradient shaping.** Given the numerous layers of

interactions that we identified experimentally, we sought for a minimal model that would capture the main mechanisms giving rise to the cellular-scale properties of the Cdc42 and Rac1 gradients. To this end, we built a one-dimensional model, where the x-axis spanned across the cell from $x = 0$ to $x = 35$ μm. We assumed that the Rho GTPases were activated and deactivated with first-order kinetics, and that levels of Rho GTPases equilibrated on a fast time scale. We assumed that the total amount of Rho GTPase $R_{tot}$ was not limiting. Eventually, we excluded diffusion and flow, such that the model was purely local. Thus, the local concentration of active Rho GTPase $R^*(x)$ at steady-state is of the form:

$$\frac{R^*(x)}{R_{tot}} = \frac{\sum_i \alpha_i [\text{GEF}]_i(x)}{\sum_i \beta_i [\text{GAP}]_i(x)}, \tag{1}$$

where $[\text{GEF}]_i(x)$ and $[\text{GAP}]_i(x)$ are the concentration profiles of GEFs and GAPs, and $\alpha_i$ and $\beta_i$ their associated effective activation and deactivation rates, which can be a function of the concentration of the Rho GTPases themselves in the case of crosstalks. From the full set of identified interactions (Fig. 4a), we could extract a minimal model explaining the formation of Cdc42 and Rac1 gradients (Fig. 4b). For Cdc42, the shape of the gradient can be simply given by an exponentially distributed GEF and uniform GAP (Fig. 4c):

$$\text{Cdc42}^*(x) \propto \frac{\alpha_C e^{-\frac{x}{\lambda}}}{\beta_C}, \tag{2}$$

where $\lambda$ is the decay length measured for Cdc42 itself (about 10 μm). Note that in the case of optogenetic activation, the optogenetic term $\alpha_{opto} e^{-\frac{x}{x_0}}$ most probably dominates the endogenous GEF activity ($\alpha_{opto} \gg \alpha_C$) such that the induced gradient follows the activating one. For Rac1, our model contains an exponentially distributed GEF of 10 μm decay length and uniform GAP, similarly to Cdc42, but also a second GAP (β2-chimaerin) exponentially distributed with its own characteristic length $\gamma = 5$ μm:

$$\text{Rac1}^*(x) \propto \frac{\alpha_R e^{-\frac{x}{\lambda}}}{\beta_R + \beta_b e^{-\frac{x}{\gamma}}}, \tag{3}$$

where $\beta_b$ is the effective rate constant for β2-chimaerin GAP activity on Rac1. This expression for Rac1 is sufficient to explain the bump (Fig. 4c), the position of which is determined by the ratio $r = \beta_R / \beta_b$ between the strength of the uniform GAP over the

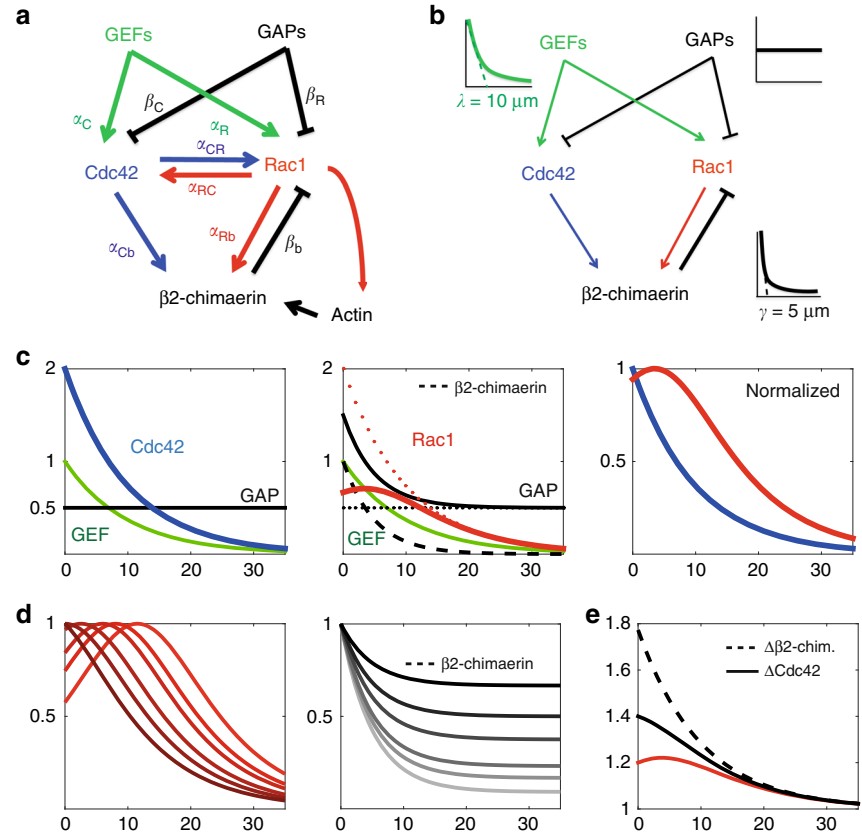

**Fig. 4** A minimal model for Cdc42 and Rac1 gradient formation. **a** Full model based on our experimental findings. Activation rates are denoted by $\alpha$ and deactivation rates by $\beta$. **b** A minimal model that recapitulates the formation of Cdc42 and Rac1 gradients. **c** Gradient shaping of Cdc42 and Rac1. Since the absolute amplitude of Rho GTPases are unknown, we assigned the following arbitrary values to the rates: $\alpha_C = \alpha_R = 1$; $\beta_C = \beta_R = 0.5$; and $\beta_b = 1$. Left: Cdc42 (blue) is set by an exponentially distributed GEF (green) with a characteristic length $\lambda = 10\,\mu m$ and uniform GAP (black). Middle: Rac1 (red) requires an additional GAP, $\beta2$-chimaerin (dashed black, characteristic length $\gamma = 5\,\mu m$), that is localized more sharply at the cell edge than Rac1 GEF (green). The overall GAP activity (plain black) is the sum of $\beta2$-chimaerin and a uniform GAP (dashed black). As a result, the putative Rac1 gradient without $\beta2$-chimaerin (dashed red) is chopped off at the cell edge resulting in a bell-shaped gradient (plain red). Right: once normalized to 1, Cdc42 and Rac1 gradients present distributions that are similar to the ones measured in cells. **d** Effect of the relative ratio $r = \beta_R/\beta_b$ between uniformly distributed GAPs and the localized gradient of $\beta2$-chimaerin on the position of the Rac1 bump. Left: Rac1 gradients obtained with decreasing values of $r$ ($r = 2, 1, 0.6, 0.3, 0.2, 0.1$ respectively from dark to light red). Right: exponentially distributed $\beta2$-chimaerin (dashed line) and uniformly distributed GAPs ($\beta_R = 2, 1, 0.6, 0.3, 0.2, 0.1$ from dark to light gray, solid lines) corresponding to the values used for the left plot. **e** Effects of Cdc42 or $\beta2$-chimaerin inhibition in silico on the Rac1 gradient ($\alpha_{C_b} = 0.4$, and $\alpha_{R_b} = 0.3$). The profiles are normalized (by the same factor) to match the FRET signal values measured experimentally (Fig. 2c, d)

strength of the localized $\beta2$-chimaerin, and by the characteristic lengths of the decaying profiles:

$$x_{\text{bump}} = \gamma \ln\left(\frac{\lambda - \gamma}{r \cdot \gamma}\right). \qquad (4)$$

From this relationship, we can see that a bump will be present if $r < (\lambda - \gamma)/\gamma$, which reduces to $r < 1$ using the experimental numbers for the decay lengths, or equivalently $\beta_R < \beta_b$. This means that the strength of the uniform GAP has to be less than the strength of $\beta2$-chimaerin to observe a Rac1 bump. The evolution of the bump position as a function of $r$ is presented in Fig. 4d. From the bump position observed in our experiment, we could predict that $\beta2$-chimaerin dominates the uniform GAPs by a factor of ~2. This minimal model for Rac1 can be refined to account for the respective roles of Cdc42 and Rac1 in mediating $\beta2$-chimaerin activity at the tip (Fig. 4e). Assuming that $\beta_b$ is a linear function of Cdc42 and Rac1 concentrations: $\beta_b = \beta_{C_b}\text{Cdc42}(x) + \beta_{R_b}\text{Rac1}(x)$, the model shows that Rac1 self-inhibition is required to account for the observed differences in the Rac1 gradient between cells depleted for Cdc42 and cells depleted for $\beta2$-chimaerin. Altogether, our minimal modeling approach suggests a simple

mechanism of distributed activators and deactivators that shape Cdc42 and Rac1 gradients such that their spatial extents are ultimately different. We thus anticipated that the spatial extent of these Rho GTPases would play a functional role.

**A controlled assay to monitor the dependence of cell migration on Rac1 and Cdc42 gradients.** We next questioned whether the different properties of Cdc42 and Rac1 gradients had an impact on migration properties. For this purpose, we imposed optogenetic gradients of ITSN or TIAM with increasing slopes (Fig. 2a). In order to control the experimental initial conditions, i.e. to prevent initial cell polarity prior to the optogenetic stimulation but still be able to monitor cell movement following it, we opted for a switchable micropatterning technique[50]. Cells were plated on round micropatterns, and would then keep an isometric shape until the surrounding repelling surface was rendered adhesive by coupling a fibronectin-mimicking chemical compound (BCN-RGD) that binds to the modified PLL-PEG repellent (APP). After addition of this reagent, cells were released from patterns and free to migrate on the coverslip (Fig. 5a, b, top row). Optogenetic stimulation with gradients of light concomitantly with the release of adhesion allowed us to study cell migration with one changing

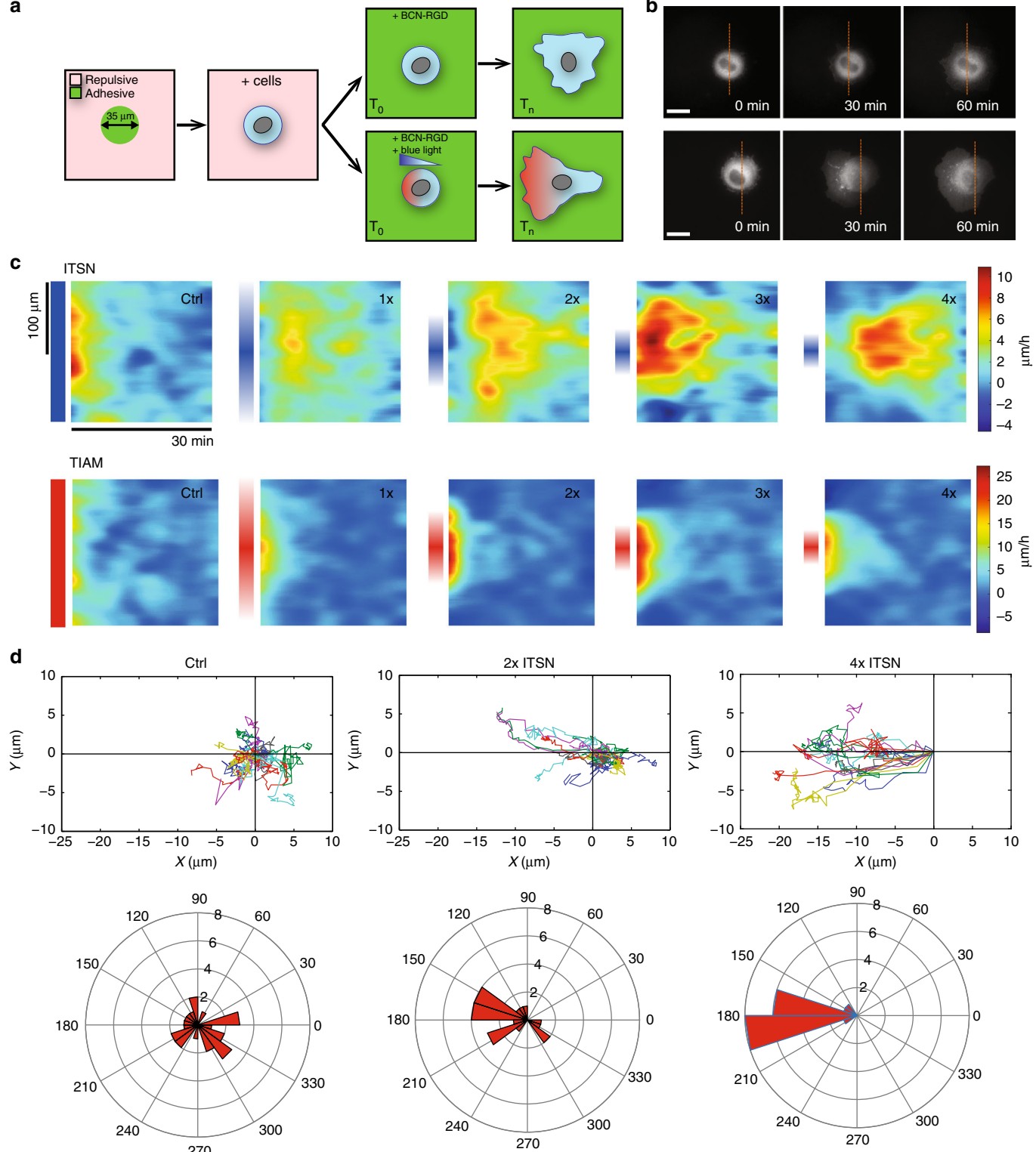

parameter, namely the extent of blue light gradients (Fig. 5a, b, bottom row). From $n \sim 20$ cells per each condition, we quantified the cell edge morphodynamics (see Methods) and averaged them for each activating gradient slope (Fig. 5c). As expected, both Rac1 and Cdc42 biased the membrane protruding activity toward the direction of the gradient. Rac1 led to an immediate cell movement while Cdc42 led to slightly delayed cell movement (Fig. 5c). We observed that cells shifted from an oriented spreading (when the back of the cell kept steady) to a directed

migration (when the back of the cell moved together with the front) by increasing the gradient slope (Supplementary Movies 3, 4). Yet, the center of mass of cells monotonously increases its movement toward the gradient as the gradient slope increased (Fig. 5d) suggesting that the quantitative properties of the gradients have a differential role in migration.

**Cdc42 provides directionality while Rac1 provides speed.** In order to assess the quantitative effect of gradient on motility, we

**Fig. 5** Scheme of the quantitative migration assay. **a** Cells are seeded on 35 µm round patterns. After complete adhesion, the adhesive reagent BCN-RGD is added and binds to the coverslip's surface, allowing free 2D cell migration (top). Directed migration can be triggered by optogenetic activation of GEFs through light gradients at the same time as cell adhesion is released (bottom). **b** Examples of cells expressing CIBN-GFP-CAAX and TIAM-CRY2-mCherry with (3× gradient, bottom) or without (top) photo-activation (visualized: TIAM-CRY2-mCherry). Time indicates the duration after addition of BCN-RGD and concomitant blue light illumination. The dashed orange line corresponds to the initial position of the cell center. **c**, **d** HeLa cells expressing CIBN-GFP-CAAX and ITSN-CRY2-mCherry or CIBN-GFP-CAAX and TIAM-CRY2-mCherry were illuminated with various gradients of light as the adhesive patterns were released. **c** Average morphodynamic maps for each condition (ITSN: top, TIAM: bottom). The vertical axis corresponds to the coordinate along the cell contour (centered on the direction of the light gradient) and the horizontal axis corresponds to time. The local velocity of the edge of the cell membrane is color coded accordingly to the bar on the right side. Gradient extents are schemed on the left side of each map. ITSN: $n=25$ (control with uniform illumination), $n=16$ (1× gradient), $n=20$ (2×), $n=19$ (3×) or $n=16$ (4×). TIAM: $n=19$ (ctrl), $n=18$ (1×), $n=18$ (2×), $n=17$ (3×), $n=18$ (4×). **d** We tracked the position of the centroid of individual cells. Top: Trajectories of cells stimulated with various gradients of ITSN. Bottom: The angles between the displacement vector (initial to final centroid position) and the stimulation axis for each cell are represented in polar coordinates. Scale bars: 20 µm

focused on two coarse-grained parameters: the maximal instantaneous velocity and the precision of the migration orientation. Sharper gradients of either ITSN or TIAM both increased cellular speed. However, activating Rac1 through TIAM had a stronger effect on speed than activating Cdc42 through ITSN (Fig. 6a), consistent with the known effect of Rac1 as a critical factor for lamellipodium formation[15] (Supplementary Figure 9). In fact, even shallow gradients of TIAM induced an enhanced migration speed. In comparison, only sharp gradients of ITSN (3× and 4×) induced an increased cellular speed, but even in these conditions speed was lower than for equivalent TIAM gradients. Conversely, ITSN gradients had a stronger effect on orientation precision. While 1× to 4× TIAM gradients had a similar effect on orientation, increasingly sharp gradients of ITSN induced an increasing precision of migration (Fig. 6b). Indeed, the 4× ITSN gradient induced the most oriented response (with a remarkable angular precision), consistent with the known role of Cdc42 as a regulator of directed migration[51,52], even though this role seems to be cell dependent[53]. We show here that directed migration is better achieved with sharp Cdc42 gradients similar to the ones measured endogenously in cells (Cdc42 gradient extent measured in migrating cells $d = 8.9 \pm 0.6$ µm, Fig. 1, 4× Cdc42 gradient extent imposed and measured through PAK-iRFP $d = 6.1 \pm 0.9$ µm, Fig. 2). Thus, in our experimental model, Cdc42 provides directionality while Rac1 provides speed of movement. These functions appear to be specific of each GTPase, since inhibition of Rac1 abolishes cell speed but not orientation for Cdc42 activation (Fig. 6c, d). Consequently, crossed activities (speed induction by Cdc42, orientation by Rac1) seem to be due to crosstalks between these Rho GTPases. Along this line, a possible functional role for β2-chimaerin is to spatially segregate Rac1 and Cdc42 activities to avoid competition between their functional roles. Indeed, as seen in supplementary figure 10, β2-chimaerin suppression has no effect on cell speed but leads to a significant reduction in angular precision. This suggests that β2-chimaerin limits Rac1 protrusive activity at the very cell front to allow Cdc42 activity to steer cell migration.

**The spatial extent but not the amplitude or slope of the Cdc42 gradient matters for directionality**. Since we showed that the shape of Rho GTPase activation gradients directly influence the outcome of cell migration, we thus questioned whether cells are actually sensitive to the slope or to the spatial extent of Rho GTPase activation gradients. In fact, in the previous experiments, both parameters varied concomitantly. It is known that cells can sense and process various extra- and intracellular signaling gradients that can hence influence cell polarity and migration[54–56]. However, it is not known to which quantitative properties of Rho GTPases intracellular signaling gradients cells are sensitive. Using the experimental setup detailed above, we could independently test the effect of gradient slope or spatial extent. When we applied

gradients of ITSN activation with different slopes but the same spatial extent, we could not detect any difference in cell motility (Fig. 6e). This also confirms that the amplitude of the imposed gradient itself does not affect the cellular response. Instead, when we imposed gradients of similar slope or amplitude but different extents, we could observe that cells stimulated with the shorter gradient of ITSN activation migrated with higher velocity and better orientation (Fig. 6e). These results indicate that the spatial extent is the critical parameter of Rho GTPase gradients read by cells.

## Discussion

In this work, we observed that the front of randomly migrating cells presents an exponentially decaying Cdc42 activity gradient whereas Rac1 shows a complex shape peaking at approximately 6 µm from the cell edge, similarly to what has been observed before in other cell types[9,36]. Combining experimental and model-based approaches, we could identify a network topology and map it spatially, allowing us to explain how these two distinct intracellular patterns are formed. By quantitatively tweaking the spatial patterns of specific GEF activity for either Cdc42 or Rac1 using optogenetics while quantifying the downstream recruitment of effectors, we showed that Cdc42 patterning can be simply explained by the combination of a localized GEF and a uniform GAP, but that Rac1 required a more complex circuitry.

We found that two mechanisms could account for Rac1 patterning. Combining one exponentially decaying GEF with either a GAP with a shorter exponential decay (like β2-chimaerin) or a directed transport from the cell front due to the actin retrograde flow was sufficient to recapitulate the observed Rac1 gradient. Yet, we showed that the effect of the actin retrograde flow does not act directly on Rac1 itself but is required for the front-most localization of β2-chimaerin. It has been previously demonstrated that the actin retrograde flow is coupled to cell polarity, by transporting various proteins away from the cell front[54]. The actin flow could act on an inhibitor of β2-chimaerin. Another possibility is that β2-chimaerin localizes at the barbed end of actin filaments thanks to its interaction with the adaptor protein Nck1[48]. Nck1 is also localized at the tip of migrating cells by the Gab1-NWASP complex[57]. Since we additionally showed that a feedback from Rac1 leads to β2-chimaerin enrichment, β2-chimaerin recruitment would depend on two concomitant signals: a Rac1-dependent signal likely going through Rac1-dependent PKC-DAG production[58], and an actin polymerizing signal through the adaptor protein Nck1. This would also explain the crosstalk from Cdc42 to β2-chimaerin through N-WASP and an increase of Nck1-mediated β2-chimaerin recruitment.

Interestingly, Cdc42 and Rac1 gradients have similar exponential decays but different spatial extents due to the local inhibition of Rac1 activity at the cell front. This observation raises important questions about the way cells interpret signaling

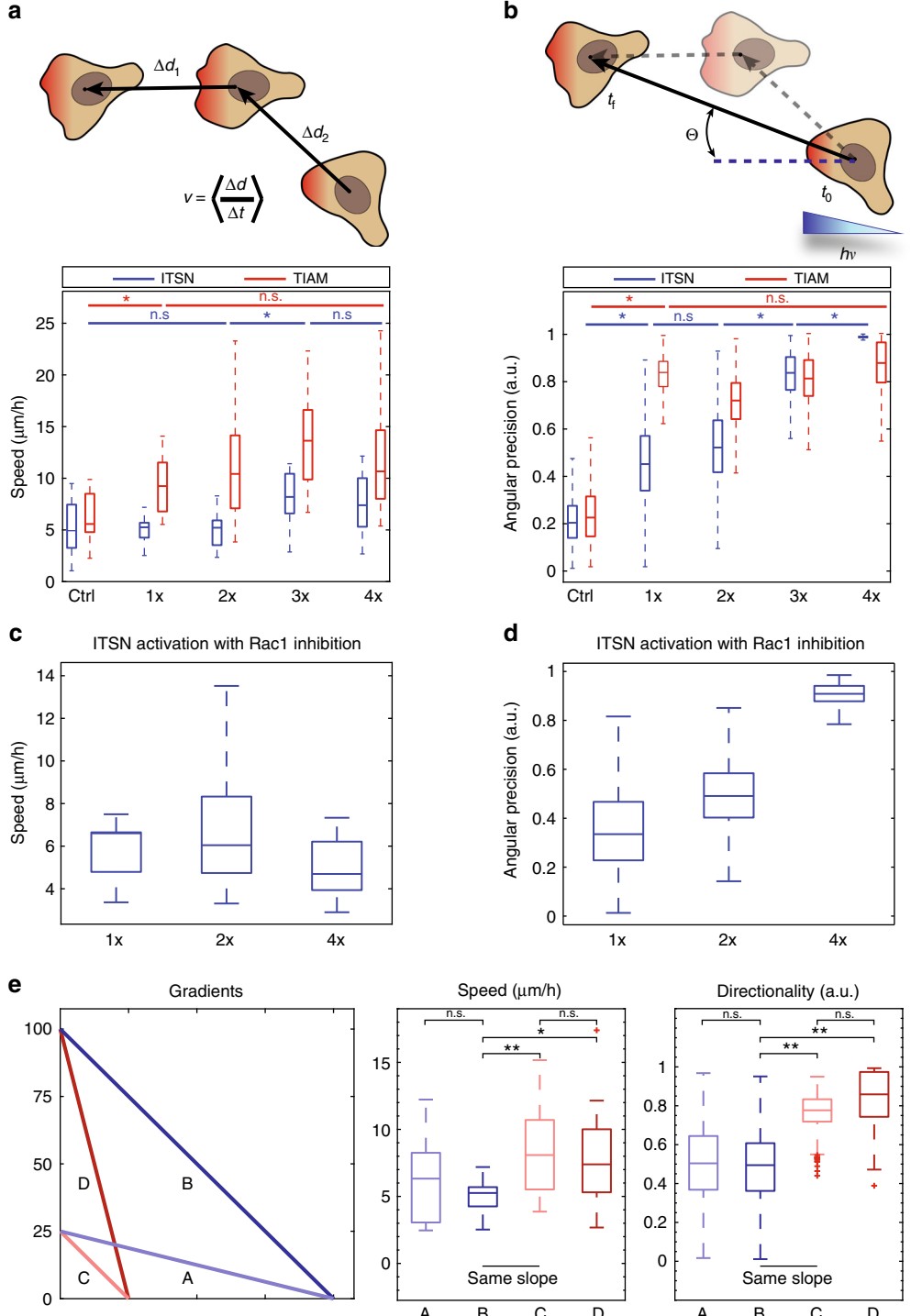

gradients. Using quantitative optogenetics, we could directly control the spatial extent, slope or amplitude of intracellular activity gradients. We showed that cell migration is not determined by the amplitude or slope of Rho GTPase gradients, but rather by their spatial extent, similarly to what was proposed in a recent work on ERK morphogen gradients in *Drosophila* embryos[59]. The spatial extent of Cdc42 needs to be small to ensure fine directionality in cell movement, in accordance with the previously shown role of Cdc42 as the primary conductor of chemotactic steering and cell polarity[9]. The spatial extent of Rac1 is larger, providing speed to the cell. Yet, since we could not effectively apply sharper Rac1 gradients without disrupting the

network topology, we do not know if the spatial extent of Rac1 presents a functional optimum as for Cdc42. Our approach can appear similar to the recent work of Zimmerman et al. who used optogenetic activations of Cdc42 and Rac1 to guide cell migration[60]. However, in their work they imposed long-range light gradients to mimic external chemo-attractant gradients, whereas in our work we imposed subcellular light gradients to keep the cell and not the environment as the relevant spatial referential of Rho GTPase gradients.

In this study, we did not consider the temporal dynamics of Rho GTPase activities. While it is very likely that spatial and temporal dynamics are connected in freely migrating cells and

**Fig. 6** Cdc42 and Rac1 drive different cellular responses. **a, b** Trajectories of cells stimulated as represented in Fig. 5 were analyzed quantitatively. **a** Cell speed defined as the instantaneous velocity of the cell displacement averaged over five consecutive time frames (top scheme). Box plots show instantaneous velocity of cells expressing CIBN-GFP-CAAX together with ITSN-CRY2-mCherry (blue) or TIAM-CRY2-mCherry (red) stimulated with various gradients of light (ITSN: n = 25 (ctrl), n = 16 (1×), n = 20 (2×), n = 19 (3×), n = 16 (4×), TIAM: n = 19 (ctrl), n = 18 (1×), n = 18 (2×), n = 17 (3×), n = 18 (4×)). Box plots represent the median, interquartile (box), 1.5 IQR (whiskers). **b** Directionality defined as the angular precision of cell displacement: the angle of displacement was measured using the initial position averaged over the first three frames and the final position averaged over the last three frames (top scheme). Angles were then bootstrapped and angular precision was calculated with the formula $r = \sqrt{(1/n^* \sum_{i=1}^{n} \sin \theta_i)^2 + (1/n^* \sum_{i=1}^{n} \cos \theta_i)^2}$. Box plots represent the median, interquartile (box), 1.5 IQR (whiskers). Statistical significance between consecutive conditions was evaluated using Wilcoxon's rank sum test. *$p \leq$ 0.05. n.s., nonsignificant ($p > 0.05$). **c** Speed and **d** directionality measurements on HeLa cells expressing CIBN-GFP-CAAX and ITSN-CRY2-mCherry after Rac1 inhibition with 100 μM NSC 23766 and stimulated with various Cdc42 gradients. n = 8 (1×), n = 16 (2×) or n = 17 (4×). **e** Effects of slope, amplitude, and spatial extent of Cdc42 gradients on cell velocity and angular precision. HeLa cells expressing CIBN-GFP-CAAX and ITSN-CRY2-mCherry were stimulated with varying gradients of light. Gradients in blue (**a, b**) and red (**c, d**) have two distinct spatial extents. Gradients in light (**a, c**) and dark (**b, d**) color have two distinct amplitudes. Two gradients (**b, c**) have the same slope. n = 13 (**a**), n = 16 (**b**), n = 22 (**c**), n = 16 (**d**). *$p \leq 0.05$, **$p \leq 0.01$, n.s., nonsignificant ($p > 0.05$) (Wilcoxon rank sum test)

while Rho GTPases patterns evolve on timescales of ~100 s[19], the response functions we measured under our steady optogenetic activations did not show evolving spatiotemporal patterns (Supplementary Figure 4). Thus, even if our synthetic approach does not recapitulate the full spatiotemporal complexity seen in native cells, we can consider our results as an example of the signaling network capacity to respond to spatially modulated inputs. Given the similarity between the native and induced gradients of Rac1/ Cdc42, we can be confident that the mechanisms we propose for gradient shaping are biologically relevant, at least at the coarse-grained cellular scale.

Following a correlative approach, Yamao et al. recently studied in time and space the patterns of Rac1 and Cdc42 activities and their link with the membrane dynamics in randomly migrating cells[61]. They concluded that Cdc42 induces random cell migration and Rac1 is responsible for persistent movement. While it might sound different from our results, the discrepancies might be explained by the scales and parameters observed in each case. We measure local and instantaneous quantities (speed and directionality), and Yamao and colleagues measure integrated and macroscopic ones (persistence and randomness). We were not able to measure those integrated quantities, since our optogenetic activations were not following the cells as they moved out of the adhesive micro-patterns. However, these different scales can be reconciled. As we conclude that Rac1 provides cells with higher speed, it also means long-term movement is more persistent[54,62]. Similarly, since we show that sharp Cdc42 gradients can fine-tune directionality, local and transient Cdc42 pulses could steer cells randomly in complex trajectories.

The minimal circuitry that we identified as sufficient to shape Cdc42 and Rac1 gradients raises new unanswered questions. In particular, it is unclear how gradients of GEFs and GAPs are shaped throughout the cell, beside the formation of the β2-chimaerin gradient we identified. Our results suggest a role for the cytoskeleton itself and its dynamics to enrich β2-chimaerin at the cell border. More generally, actin networks and actin-regulating complexes can act as scaffolding complexes in protrusive regions where they localize. For example, the WAVE Complex, a downstream effector of Rac1, recruits WRP, a GAP inhibiting Rac1[63]. Similarly, N-WASP, a downstream effector of Cdc42, associates with the GEF ITSN. More mechanisms are probably involved. In particular, membranes could play a direct role in the localization of these regulators of Rho GTPase activity. The local lipid composition, and in particular the concentration of PIP3, has been shown to control the activity of Rac1 and Cdc42[26,28,64]. In addition, membrane curvature-sensing BAR proteins localize at highly bent membranes, including cell edges. Several BAR proteins are known to bind Rho GTPases or their regulators. IRSp53, a member of the I-BAR family found in

lamellipodia and filopodia has been shown to bind Cdc42, Rac1 and WAVE2[65,66]. Even if β2-chimaerin was sufficient to explain Rac1 shaping in the present work, other known GAPs, such as ARHGAP22, ARHGAP24 (FILGAP) and SH3BP1, interact with the proteins involved in cell protrusion and could play a similar role as β2-chimaerin. In particular, it was previously shown that depletion of SH3BP1 results in a high activity of Rac1 at the front[67]. Also, it remains to be explored if the Cdc42 and Rac1 positive feedbacks and crosstalks, as previously suggested[9] and observed in our work (Fig. 3b), play a role in shaping GEF distributions.

## Methods

**Plasmids and molecular constructs**. ITSN-CRY2-mCherry was constructed as detailed previously[44]. The TIAM DH-PH domain was similarly amplified from TIAM(DHPH)-Linker-YFP-PIF (gift from O. Weiner, University of California, San Francisco) and cloned into CRY2PHR-mCherry. Both ITSN-CRY2-mCherry and TIAM-CRY2-mCherry were cloned in pHR lentiviral vectors (gift from O. Weiner) by Genscript (Nanjing, China) using *Mlu*I and *Bst*BI cloning sites. N-WASP-iRFP, PAK1-iRFP and β2-chimaerin-iRFP fusion genes were constructed by Genscript (Nanjing, China) by cloning the corresponding human cDNAs upstream the iRFP713 gene sequence[68], separated by a PVAT sequencer. The Abi1-iRFP plasmid was kindly provided by Maria Carla Parrini. Rac1BS and Cdc42BS plasmids were kindly provided by Dr. Louis Hodgson[35], and were subcloned into the lentiviral pLVX vector (Clontech, Mountain View, CA USA) between *Xma*I and *Xba*I cloning sites.

**Cell culture and reagents**. HeLa cells (CCL-2 strain, bought from ATCC) were cultured at 37 °C with 5% $CO_2$ in Dulbecco's modified Eagle's medium supplemented with 10% fetal bovine serum and penicillin-streptomycin (100 U/mL). Transfections were performed using X-tremeGENE 9 (Roche Applied Science, Penzburg, Bavaria, Germany) according to the manufacturer's instructions using an equal amount of plasmid DNA for each construct (1 μg). Stable cell lines were obtained using lentiviral infections: all lentiviruses were produced by transfecting pHR- or pLVX-based plasmids along with the vectors encoding packaging proteins (pMD2.G and psPax2) using HEK-293T cells. Viral supernatants were collected 2 days after transfection and HeLa cells were transduced at an MOI of 2. Gene expression knockdown was achieved using pooled siRNA with the following sequences. Cdc42: 5′- CGAUGGUGCUGUUGGUAAA-3′ and 5′-CUAUGCAG UCACAGUUAUG-3′, β2-chimaerin: 5′- AUUGAAGCAAGAGGAUUAA-3′ and 5′-CCACUUCAAUUAUGAGAAG-3′, Rac1: 5′-UUUACCUACAGCUCCGU CUUU-3′ and 5′-UACAGCACCAAUCUCCUUAUU-3′, ctrl: 5′-AGGUAGU GUAAUCGGCCUUG-3′ and 5′-GCGGGAUAUUUCGGUCAAU-3′. siRNA transfection was done following the manufacturer's protocol (Lipofectamine RNAiMax, Thermo Fischer Scientific), and cells were imaged 48 h after transfection. The JLY cocktail (8 μM jasplakinolide, 5 μM Latrunculin B, 20 μM Y27632) was applied 15 min before image acquisition.

**Live cell imaging and optogenetics**. Micropatterned coverslips were prepared as described by Azioune et al.[69]: $O_2$ plasma-cleaned coverslips were incubated with 0.1 mg/ml of PLL-g-PEG (Surface Solutions, Switzerland) in 10 mM HEPES, pH 7.4 for 1 h. They were then exposed to deep UV through micropatterned quartz/ chrome photomasks (Toppan, Round Rock, TX) for 5 min, and incubated with fibronectin in 100 mM $NaHCO_3$ (pH 8.5) for 1 h. Releasable micropatterns were prepared similarly, with PLL-PEG being replaced by azido-PLL-g-PEG (APP) at 100 μg/ml. Migration was released by addition of 20 μM BCN-RGD for 10 min. Before imaging, cells were dissociated using Versene (Life Technologies) and

seeded for adhesion on the previously mentioned coverslips for at least 2 h. Experiments were performed at 37 °C in 5% $CO_2$ in a heating chamber (Pecon, Meyer Instruments, Houston, TX) placed on an inverted microscope model No. IX71 equipped with a ×60 objective with NA 1.45 (Olympus, Melville, NY) and a Luca R camera (Andor, Belfast, UK). The microscope was controlled with the Metamorph software (Molecular Devices, Eugene, OR). TIRF images were acquired using an azimuthal TIRF module (iLas2; Roper Scientific, Tucson, AZ). Optogenetics stimulations were performed every 30−40 s with a DMD in epi-mode (DLP Light Crafter, Texas Instruments) illuminated with a SPECTRA Light Engine (Lumencor, Beaverton, OR USA) at 440 ± 10 nm.

**FRET**. HeLa cells were lentivirus-infected with a Cdc42-FRET-biosensor or a Rac1-FRET-biosensor (kindly provided by Louis Hodgson) and sorted for intermediary fluorescence using fluorescence-activated cell sorting (FACS). Twenty-four hours after plating them on glass coverslips, cells were imaged by TIRF microscopy. Excitation was done with a laser at 405 nm, dichroic mirrors stayed the same (BS: FF-458-DiO2, Semrock) while a filterwheel allowed for the switching of appropriate emission filters to acquire sequentially donor (mCerulean, Em: FF01-483/32) and FRET (Em: FF01-542/27) emissions. Image processing included registration, flat-field correction, background subtraction, segmentation, and FRET/donor ratio calculations. FRET profiles measured from the FRET images were normalized between 0 and 1 when comparing the two Cdc42 and Rac1 FRET reporters or when comparing a FRET reporter with another fluorescence signal. We did not normalize FRET profiles from the same reporter when comparing two different experimental conditions.

**Immunofluorescence microscopy**. HeLa cells stably expressing a Rac1-FRET-biosensor were fixed in phosphate buffer saline (PBS) with 4% paraformaldehyde for 20 min at 25 °C. After permeabilization in PBS + 0.1% triton X-100 for 15 min and blocking in PBS + 1% BSA + 1% FBS for 20 min, stainings were performed in PBS with 0.05% triton + 1% BSA 1 h at room temperature. Antibodies were used as follows: β2-chimaerin primary antibody: 1 /100 (Orb182594, Biorbyt), anti-rabbit-Alexa594 antibody: 1 /400 (ThermoFisher). Phalloidin-488 was used at 300 nM. Acquisitions were made in HiLo mode using an azimuthal TIRF module as described above).

**Image processing and quantification of intracellular gradients**. Images were analyzed with custom-built Matlab routines. For the images obtained in our optogenetic experiments, we subtracted the initial pre-optogenetics signal from all subsequent images in order to measure solely the recruitment of fluorescent proteins to the basal membrane and to avoid volume artifacts. The resulting differential images were normalized between 0 and 1 using the same normalizing factors as the gradients quantified from each image (see below). Normalized images were then averaged over ten time points and over all cells to produce the averaged images shown in Figs. 2d, f and 3b, d. The associated Fire color scale was defined as the average temporal fold change of fluorescence that we measured from the raw images after background subtraction (taken outside the cell mask). For the quantification of the gradients presented in Figs. 1b, and 3c, f, g, we measured the FRET ratio along two linescans per cell, drawn manually perpendicular to the cell edge in protrusive regions with a line width of 10 pixels. The gradients were first averaged for each cell, and then averaged over all cells. For the quantification of the gradients presented in Figs. 2b, e, g and 3e, h, i, fluorescence was quantified along a line of 10 pixels in width spanning across the cell diameter in the direction of the optogenetic gradients. The curves in Figs. 2b, e, g and 3e, h, i were normalized between 0 and 1 where 0 stands for the average of the five minimal values and 1 stands for the average of the five maximal fluorescence values.

**Processing of the migration movies**. Movies were analyzed with custom-built Matlab routines. The segmentation of cell borders was performed on fluorescence images using the Matlab function Graythresh. Cell centroid positions were determined using the Matlab function Regionprops and used to quantify cell movement. To measure cell velocity, we computed instantaneous speed of cell centroids at each time frame, and then averaged it over several time frames. Cells stimulated through TIAM activation reached maximum speed soon after the beginning of illumination, so instantaneous speed was averaged between $t = 15$ min to $t = 45$ min. Cells stimulated through ITSN activation reached maximum speed at later stages, and instantaneous speed was thus averaged between $t = 60$ min to $t = 90$ min. Angular precision was computed as follows: for each cell, the displacement vector was computed between the initial cell centroid (averaged over the three first time frames) and the final cell centroid (averaged over the three last time frames), and we measured the angle between this vector and the axis of stimulation gradients. These angles were bootstrapped over 1000 replications, and angular precision was estimated with the formula

$$p = \sqrt{\left( (1/n^* \sum_{i=1}^{n} \sin \theta_i )^2 + (1/n^* \sum_{i=1}^{n} \cos \theta_i )^2 \right)}.$$ The morphodynamics maps (Fig. 5c) were obtained using a routine adapted from Yang[9]. The cell contour was aligned such that the middle of the map was centered on the direction imposed by the optogenetic gradient.

## Data availability

The data that support the findings of this study and all custom codes used for analysis are available from the corresponding author upon reasonable request.

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

## Acknowledgements

We thank Louis Hodgson for sharing the Rho GTPase sensors; Maria Carla Parrini, Kristine Schauer, and Christophe Tribet for critical reading of the manuscript and helpful discussions; Remy Fert and Eric Nicolau for their technical assistance; The Institut Curie Shared FACS Facility for cell sorting. M.C. acknowledges financial support from French National Research Agency (LICOP no. ANR-12-JSV5-0002-01). M.C. and M.D. acknowledge funding from French National Research Agency (ANR) Paris-Science-Lettres Program (ANR-10-IDEX-0001-02 PSL), Labex CelTisPhyBio (No. ANR-10-LBX-0038), the France-BioImaging infrastructure supported by ANR Grant ANR-10-INSB-04 (Investments for the Future), and Institut Pierre-Gilles de Gennes (laboratoire d'excellence, Investissements d'Avenir program ANR-10-IDEX-0001-02 PSL and ANR-10-LABX-31).

## Author contributions

Conceptualization, S.D.B., M.D. and M.C.; Methodology, S.D.B., M.D. and M.C.; Software, S.D.B. and M.C.; Formal analysis, S.D.B., K.V. and M.C.; Investigation S.D.B., K.V. and J.M.; Resources, F.dF., G.C., and F.D.; Writing—original draft, S.D.B. and M.C.; Writing—review and editing, S.D.B., M.C., M.D. and K.V.; Visualization, S.D.B. and K.V.; Supervision, M.C.; Funding acquisition, M.C.

## Additional information

**Competing interests:** The authors declare no competing interests.

