## [Peer Review File · Nature Communications]

Editorial Note: This manuscript has been previously reviewed at another journal that is not operating a transparent peer review scheme. This document only contains reviewer comments and rebuttal letters for versions considered at Nature Communications

Reviewers' comments:

Reviewer #1 (Remarks to the Author):

I thank the authors for revising their manuscript and addressing many of the points raised by me and the other referees. This is a large body of work, which should be of wide interest to those studying cell migration, the role of Rho GTPases in cell migration, as well as the formation and regulation of intracellular signaling gradients. Even though I suspect that other GEFs and GAPs will be important, I accept that pursuing all of these is beyond the scope of the current study. In its current form, the work advances the field. They use innovative technical approaches and these will also be of interest to cell biologists and others in this field.

Reviewer #2 (Remarks to the Author):

The authors have addressed most of my technical and other concerns except of providing more of the requested mechanistic and molecular insights into the regulation of the Rac and Cdc42 gradients (points 1 and 2). This reduces my enthusiasm for an otherwise interesting study both in terms of their methodology and also of the discussed ideas.

Reviewer #3 (Remarks to the Author):

Review report

Titel: Optogenetic dissection of Rac1 and Cdc42 gradient shaping

The manuscript by De Beco and colleagues describes an optogenetic approach in combination with micro-patterning to characterize Cdc42 and Rac1 activation gradients in HeLa cells. Using optogenetic activation of RhoGEFs in a confined part of the cell, they show that Rac1 and Cdc42 activity gradients show an exponential decay towards the center of the cell. The Cdc42 activity patterns can be explained by the GEF activity patterns that the authors impose on the cell, combined with global GAP activity. Following the observation that Rac1 activity peaks around 6µm from the cell edge after similar GEF stimulation, the authors propose a model where Rac1 is deactivated at the cell edge by a specific GAP, beta2-chimaerin, on top of global deactivating GAP activity. This specific Rac activation pattern is influenced by Cdc42 expression levels and actin dynamics based on siRNA and drug perturbations. Based on a mobility assay to quantify cell polarization parameters and velocity, the authors show that smaller Cdc42 activation patterns cause higher polarization precision, while the size of activity patterns for Rac1 does not seem to have an influence on velocity. They propose that the width of the activation pattern of the Rho GTPases is a crucial factor influencing velocity and polarization, as opposed to amplitude/slope of the activity pattern.

Although synthetic, the approach the authors use nicely separates mechanical influence of polarisation and cell to cell heterogeneity in shape from internal signaling gradients, thus allowing a more quantitative comparison between different experimental conditions.

While the revised manuscript is improved in terms of data presentation and clarity, there are still some major technical and mechanistic questions to be addressed. Reading the revised manuscript also raised some new concerns, mainly around the data presented in figure 3, please see below.

Major points:

-Figure 3b, there seems to be a difference in the average recruitment pattern of Abi1 between Tiam and ITSN, where the Tiam mediated recruitment is sharper PM localised, the ITSN pattern seems to have a wider extent into the cell. The authors state in the accompanying text that both are similarly located at the plasma membrane, but this is obvious from the raw data and not quantified. A quantification of this difference should be performed and commented on in relation to the conclusions on crosstalk between Cdc42/Rac1?

-Figure 3: The data with the use of Abi1 as read-out for Rac1 activity is contradictory to the Pak1/FRET read-outs.

One of the main claims of the authors in the manuscript is that Rac1 has a distinct activation pattern peaking 6µm from the cell edge (based on Rac FRET/Pak1 effector imaging read-outs). However, when they used a specific effector for Rac1, Abi1, they find that activated Rac1 has a peak at the cell edge upon activation of the same GEF?? How do the authors reconcile these apparently contradicting results, e.g. 2f vs 3b?

-Figure 3c: Here the authors show that Rac1 activity measured by FRET forms the same distinct pattern with 6µm peak activation (red curve) WITHOUT any optogenetic stimulation of GEFs, e.g. 2g vs 3c.

In Suppl. Figure 1a, the authors show Pak1 activity without optogenetic stimulation. Quantifying the fluorescence intensity plot in this figure would help interpretation and strengthen the comparability of the different read-outs for Rac1 activity used between figures. Also would be helpful to see the same control for Abi1. How is this read-out localised with only control CRY2-mCherry stimulation?

-Figure 3, 3c, f: Along the same lines as above, showing the ctrl siRNA versus the Cdc42 siRNA/beta2-Chimaerin siRNA on the PAK1-iRFP readout in unstimulated cells would greatly improve credibility of the crosstalk points in the manuscript and prove consistency between the different read-outs for Rac1 activity. I suggest adding this data, because right now it seems the FRET read-out shows differential activation patterns from PAK1. Which effector domain is in the FRET biosensor for Rac1, is this also PAK1 or another effector domain? This is unclear from citation.

-In the introduction the authors claim "We show that the localized activity of β2-chimaerin depends on a negative feedback from both Cdc42 and Rac1". This is a confusing statement without any data in the manuscript to back this claim up, how do the authors investigate negative feedback from Cdc42 and Rac1 on beta2-chimaerin here?

-The authors claim that "that the actin retrograde flow is required for β2-chimaerin enrichment", but should do additional experiments to back this statement up (e.g. show co-localisation of beta2-Chimaerin with F-actin) or precise their conclusions. The use of the JLY cocktail is not controlled now, because it is not specific and potentially effects many proteins and processes involved in the RhoGTPase/cytoskeletal machinery. E.g. can the authors show one protein that does stay localised to the plasma membrane after this drug is added?

-Suppl. Figure 1b: Based on the average iRFP localisation (13 cells), it seems really localised to the top of these cells. This raises the question whether the iRFP TIRFM illumination is even across the field of view, did the authors control/correct for this, as this could potentially be a major confound in the interpretation of the major iRFP-fused read-outs in the study?

-Adding a scalebar to averaged intensity images (mCherry/iRFP) would help the reader with interpreting the accompanying intensity plots (or even better draw the lines where the plots are quantified from). It is confusing for the reader as they sometimes seem to contradict while it

should be showing the same data:

For example the quantification in 3d,e; The averaged beta2-Chimaerin fluorescent extend seems to stay higher further into the cell in the ITSN condition in comparison with the TIAM condition (3d, based on 0-1 color bar), but the quantification in 3e shows that in the TIAM condition the fluorescence is higher at almost every point along the lineplot?

-Figure 6: The terminology used to describe the results in this section and the figure legends is very confusing and ambiguous.

How are "slope", "spatial extent" and "amplitude" defined here and related to each other? As far as I can see, the authors use "slope" merely as a function solely dependent on both spatial extent and amplitude, so is unnecessary to name and visualise separately.

The only thing the authors experimentally test here is the influence of "amplitude" (meaning light intensity of CRY2-CIBN dimerization signal, providing information about when the dimerization system is saturated), and the influence of "spatial extent" (providing information about spatially restricting activation of the Rho GTPase in the cell). Figure 6e and 6f provide all the information necessary for interpretation and comparison already, and could even be merged for clarification...6g and 6h are just repetitions of exactly the same information and data. I would suggest for the sake of interpretation and clarification that the authors remove the word "slope" from the manuscript and simplify their descriptions of the data and figures.

Minor points:

-The term "advection" should be replaced by "retrograde actin flow" or be more precisely defined in context, as also suggested by another reviewer.

Currently it is confusing for readers and the new definition the authors provide "...advection refers to the oriented (ballistic??) transport of material.." only makes interpretation more ambiguous.

-Is the DMD mediated "activation" light sheet that is produced also in TIRF modus?

-Figure 1b, are mean or median curves plotted here? (description also missing in legends of some other plots)

-P20L18-20: "Combining one exponentially decaying GEF with either a GAP with a shorter exponential decay (like β 2-chimaerin) or advection from the cell front is sufficient to recapitulate the observed Rac1 gradient"

What do the authors mean with this statement? Advection of what from the cell front?? Please rephrase.

Textual:

P2L29: "...Fibroblats.." should be corrected to "Fibroblasts"

P2L45: "...anther.." should be corrected to "another"

P3L12: suggest to remove "classes of" to improve interpretation

P5L12: suggest to remove "...futile.." here to improve interpretation

P5L12: "switch" should be corrected to "switches"

P5L14: "Two limit cases can be envisaged" would advise to rephrase for improved interpretation.

P5L23: "model" should be corrected to "models"

P3L31-33: suggest to add "activity" before Rac1 here, to improve interpretation

P4L21: "model" should be corrected to "models"

P6L18: "ouput" should be corrected to "output"

P6L30: suggest to replace "observe" with "show" to improve interpretation.

P6L37: "bounded" should be corrected to "bound"

P21L21: suggest to replace "a signature" with "an example" to improve interpretation.

P21L24: suggest to replace "significant" with "relevant" to improve interpretation.

Based on the above, I cannot recommend this study for publication now. Some technical and mechanistic details and controls are still missing or unclear, and this hampers the evidence for the biological interpretation of the main claims involving beta2-Chimaerin that the authors make. A thorough revision with added experimental data and improved data and textual presentation would potentially render this manuscript suitable for publication in Nature Communications, as the approach and results would be of interest for the field.

Reviewer #1 (Remarks to the Author):

In this manuscript De Beco et al. investigate the basis for the shapes of gradients of Rac1 and Cdc42 in migrating cells. The authors use optogenetics to locally activate the GEFs Tiam and Intersectin (ITSN) to artificially generate active Rac1 or Cdc42 gradients, respectively. In order to prevent cell polarity arising from cell shape to influence the induced gradient of activated Cdc42 or Rac1 they use circular micropatterned substrates to confine the cells they are studying. Active Cdc42 showed an exponentially decaying gradient from the protruding cell edge (or site of ITSN activation) going back toward the nucleus. In contrast, the Rac1 gradient peaked a short distance behind the leading edge (consistent with previously published work from others) and then decayed less steeply than the Cdc42 gradient. A major goal of this study is to determine what accounts for the pattern of these gradients. Is it due to a “synthesis, diffusion, degradation” pathway or is it due to a gradient of activators and deactivators (i.e. GEFs and GAPs)?

With Cdc42, they find that the pattern of activation of the GEF determines the pattern of the gradient. Diffusion or transport are not involved. However, the origin of the Rac gradient is more complex. They ask whether it is due to transport or to non-uniform distribution of deactivators (GAPs). The authors provide evidence that the localization of a specific GAP, β 2-chimaerin, at the leading edge is critical for the Rac1 gradient peaking a short distance behind the leading edge. Evidence is presented that recruitment of β 2-chimaerin to this location depends on retrograde actin flow and that when this is blocked not only is there loss of β 2-chimaerin in this region but that the Rac1 activity gradient becomes more uniform and reflects the pattern of GEF activation. These are interesting results and important conclusions.

Additionally, some of the most interesting experiments are those examining the effect of the shape, amplitude and spatial extent of the gradients on cell migration rate and polarization (figure 6). These gave unexpected results (for me, at least) and consequently I found these particularly interesting and worthy of publication.

We thank Reviewer 1 for his/her comments and interest for our work. Bellow, we answer point by point to his/her suggestions.

However, I do have some specific comments that need to be addressed.

Comments

1. Much of the work in the paper focuses on the GAP β 2-chimaerin and its contribution to the pattern of Rac1 activation, giving rise to the decrease at the leading edge or site where Tiam is activated. The authors seem to have picked β 2-chimaerin because previous work has shown it to be localized to the leading edge of cells. However, there are many Rac GAPs and most cell types express multiple GAPs for each Rho family GTPase. Several Rac GAPs have been localized to the leading edge of cells. These include ARHGAP22 and ARHGAP24 (FilGap), as well as srGAP2. Are any of these other GAPs expressed in the authors' system? They need to examine this and to determine whether any of these similarly function like β 2-chimaerin. Notably, srGAP2 contains a BAR domain and so this might be localized to the leading edge by interaction with the curved membrane surface.

We agree with Reviewer 1, there are many GAPs known for Rac1, and several GAPs that localize to the cell edge beyond β 2-chimaerin (such as srGAP2, IRSp53, or SH3BP1). Yet, given that we observed the disappearance of the Rac1 “bump” with siRNA against β 2-chimaerin (Fig 3f), we concluded that β 2-chimaerin was sufficient to understand the mechanism by which Rac1 was shaped. Our results do not exclude supplemental layers of regulations or redundant mechanisms mediated by other GAPs but we thought that an exhaustive analysis of all Rac1 GAPs was beyond the scope of our work.

Nevertheless, we extended our paragraph in the discussion on the other Rac1 GAPs and we suggested that the understanding of their specific role would be a topic of interest for further studies (page 22 lines 10-19).

2. Not only do the authors need to examine additional Rac GAPs but they also need to perform some knockdown/rescue experiments with β 2-chimaerin. When it is knocked down, is its effects on the Rac1 gradient rescued by expression of wildtype β 2-chimaerin, GAP-deleted β 2-chimaerin and β 2-chimaerin unable to localize to the leading edge? Results from these experiments should strengthen or weaken their arguments.

We agree with reviewer 1 that his/her suggested rescue experiments with mutants would refine our argument. Yet, these experiments are not easy to perform since we would need to design β 2-chimaerin mutants that would not be targeted by siRNAs. Even if we use synonymous mutation on the β 2-chimaerin sequence, we might encounter issue with expression levels (even the wildtype β 2-chimaerin fused to iRFP was hard to express). Future work using genome-editing techniques such as CRISP/CAS9 would be more suited for this purpose.

3. This paper is well written but for a paper in NCB it is unnecessarily wordy. A lot of the introduction could be shortened.

We shortened and rewritten parts of the introduction.

Minor points:

1. What is the cocktail "JLY"? A reference is given but most people won't take the time to look this up. It would be very easy to indicate the composition in parentheses. I guessed correctly what this indicated but I had to go to the reference to confirm my guess.

We apologize for using the JLY acronym without definition. We now defined it page 9 line 5.

2. In the references, multiple journal names are omitted, e.g. for references 8, 9, 13, 18, 20, 21, 26, etc.

We corrected these omissions.

3. "Advection" is a term I have rarely if ever heard in the context of cell biology. I realize that they are using it to refer to transport and specifically here to retrograde actin flow. I would suggest that they refer directly to retrograde actin flow and avoid any confusion with the word "advection".

We apologize for the use of this word that appeared unfamiliar to reviewer 1. Yet, it is a common word in physics and hydrodynamics so we chose to keep it in the text as we considered it equivalent to the term "diffusion" which is well accepted in the cell biology community. To help readers, we now define it in the introduction (page 3 line 11).

Reviewer #2 (Remarks to the Author):

In this manuscript entitled “Optogenetic dissection of Rac1 and Cdc42 gradient shaping” the authors combine optogenetic, micropatterning, advanced quantitative imaging and modeling to investigate how cells polarize via spatial organization of RhoGTPase activities. They conclude that Cdc42 activity gradients are determined by a gradient distribution of its GEFs and not by GAPs. More importantly, they propose that Rac1 gradients are a result of the distributions of both its GEFs and GAPs, particularly TIAM1 and β 2-chimaerin. One of the findings of this study is that the slope and magnitude of GEF gradients have no impact on Rac1 or Cdc42 activity gradients, but rather it is likely the spatial extent of these GEF distributions that is important for speed. Crosstalk between Rac1 and Cdc42 signaling networks has been shown previously and can complicate interpretations, this study is no exception and finds that Cdc42 and Rac1 activities can alter β 2-chimaerin to lamellipodial protrusions. They propose an interesting model where β -chimaerin localization can explain a lower broadened region of Rac in the front. Overall, the study uses timely methodologies and the results might be of interest to the readership of Nature Cell Biology. However, the study appears to be preliminary with the data and controls included. There are also several key molecular/mechanistic parts missing needed for a comprehensive study. If this additional data could be added, the manuscript should be considered.

We thank Reviewer 2 for his/her interest in our work and for the numerous appropriate suggestions. We recognize that several controls and clarifications were lacking in the first manuscript that we have now addressed. Regarding the molecular mechanisms, we think that our work provides a systems-level analysis where the global properties we identified do not depend on the actual molecular mechanisms. This is the strength of our engineered approach, but also its limitation. In addition, we feel that our work brings substantial new results and concepts for the field in its present form so that the dissection of the molecular mechanisms appeared to us out of the scope of the present manuscript.

1) One main claim the manuscript is that β 2 chimaerin is regulated by feedback from Cdc42, creating a crosstalk that may explain how Cdc42 can regulate Rac at the leading edge. The data supporting this regulatory mechanism is mostly indirect and they discuss different possibilities how this may occur. Since this is an important claim of the study that is interesting if correct, they need to add additional more mechanistic data showing how Cdc42 regulates β 2-chimaerin.

We observed that i) silencing β 2-chimaerin resulted in the loss of Rac1 “bump”, ii) activating Cdc42 resulted in increased β 2-chimaerin concentration at the cell edge, and iii) silencing Cdc42 resulted in the loss of Rac1 “bump”. Even if when taken apart these are indirect evidences, we think that altogether they provide a solid argument to say that Cdc42 to β 2-chimaerin feedback is necessary for the formation of the wild-type Rac1 distribution. Yet, as reviewer 2 says, we still miss the molecular mechanism by which Cdc42 and β 2-chimaerin interact. Even if we discuss in detail a possible scenario in the discussion (page 20 lines 27-35), we think that the dissection of the precise mechanism is beyond the scope of our study.

2) Another main claim of the manuscript is a marked difference in GAP-GEF gradients for CDC42 and RAC that must be based on differential distributions of GEFs and GAPs. They conclude that CDC42 relies only on a GEF gradient alone. Since many researchers are interested in CDC42 regulation, this would be an important contribution. However, the data for this conclusion is indirect and mostly based on modeling assumptions that I think could be easily changed. Since this is a critical point of how they think the polarity system works, they need to show whether relevant CDC42 GEFs or GAPs

actually show such gradients or no gradients. Alternatively, they need to find a way to show that CDC42 inactivation rates are the same in the front and back as their model of constant GAP activity argues. Without more supportive data, the inclusion of this uniform GAP model may add more confusion to the field.

We agree with reviewer 2, in principle it would be very informative to image and quantify the Cdc42 GEFs and GAPs to show that they are distributed as we predict. However, this is a very difficult task for two main reasons. First, there is a multitude of GEFs and GAPs, and we would need an exhaustive study to confirm the distribution of each of them, which is clearly out of our scope. Actually, this is why we went for a synthetic perturbative approach: by imposing an optogenetic GEF gradient, we could reveal the overall deactivating activity resulting from all GAPs of Cdc42. If Cdc42 GAP activity was not uniform at the cellular scale, we would have observed a non-linear response at the level of the effector, as explained in the text (which shows that Cdc42 overall deactivation is the same at the front and the back). Second, the imaging and quantification of a given GEF or GAP distribution at the plasma membrane is very difficult since most of these regulators are also strongly associated with endomembranes. For example, the quantification of β 2-chimaerin in fig 3g required a gamma correction on the image to avoid saturation due to the strong signal from the ER/Golgi. In addition to antibody selectivity, which can always be questioned, these imaging limitations were preventing us to go in this direction. We chose to leave this identification of GEFs and GAPs as an open problem for further works.

We would also like to point to Reviewer 2 that we do not claim that there is just one GEF for Cdc42, we simply state that one GEF (exponentially distributed) would be sufficient. We do not exclude that a combination of GEFs could sum into an effective exponential activation profile. We modified the sentence at the end of the discussion (page 22 line 19-22) to precise this fact.

3) Many of the authors conclusions are drawn from comparisons of intensity profile distribution plots (either FRET ratios or fluorescence). A main problem is that their analysis shows that there is significant overlap between mean and SD values in most of these profiles. In many cases they note differences when the error regions show significant overlap (for examples see 1b, 2e and 2g, 3c, 3h). The authors need to add a careful statistical analysis that their conclusions about different profiles are indeed significant. In several cases, eg Fig. 3h, whether these differences are significant is absolutely crucial for their arguments and models.

Indeed a statistical measurement of profile similarity was missing in the first manuscript. We thus sought for a method to compare our measured profiles. We did not find a general statistical method that could apply to our data, so we went for a simple and comprehensive measurement. We decided to perform standard U-test (Wilcoxon rank sum) for each position of space and added on top of each figure a bar showing the portion of space over which the two profiles are statistically different. The new figures 1, 2 and 3 are presented in the revised version of the manuscript.

4) The known slow off rates for the Cry2 systems is a major problem for their analysis as well as interpretation and they need to clearly account for this problem. They need a paragraph explaining this kinetic problem of the Cry system and the resulting limitations of what they can learn from their experiments and analysis given these temporal limitations. Specifically, they need to show time courses of the kinetics of Cry2 and Pak1 recruitment while light is being applied, and they have to show how long it takes for the signals to return to normal after the system is shut off is important. For example, with such long times for activation and dissociation times it is likely that the activating

GEFs are induced much more slowly than normal and they are creating stable signaling states long after cry2 is no longer activated. Given these long times, one could argue that no strong arguments can be made about diffusion of the small GTPases and certain questions about feedback cannot be addressed with this approach. There is for example the issue that cells normally extend lamellipodia rapidly in seconds and the cells they show achieve maximal speed after ITSN activation only 45-60 minutes post illumination. I believe that this turns out to be a major problem for their study given that local CDC42 and Rac activation can trigger actin polymerization and membrane protrusion more than a 100-times faster (on the time scales of a few seconds in other reported experiments during membrane protrusion rather than an hour). They need to clearly address these limitations of their approach and clarify which conclusions they can make and which ones they cannot make with this approach.

Regarding the first part of this comment, we disagree with Reviewer 2. The slow off rate (~180s) of the Cry2 system is not a limitation for the present study. As we have shown in a previous publication (Valon et al, *Biophys. J.* 2015) the activation is fast (it takes ~3s for Cry2 to bind on the plasma membrane). The dissociation is indeed slow, but since we are interested in steady state and not by signal decays, it is not a limitation. We already showed the time courses of Cry2 and Pak1 recruitment in supplemental figure 5. The 45-60 minute delay in migration is due to the APP/BCN-RGD surface chemistry and not to the optogenetic system since we observed that, on normal glass, protrusions are induced within 2-3 minutes upon light illumination.

Yet, Reviewer 2 is right regarding the fact that Cdc42 and Rac1 patterns of activity are evolving on timescales of ~100s in native cells. However, as stated in the discussion, we only considered here the spatial properties of the signaling network and not its temporal characteristics. We showed that biologically significant regulation in the network could be revealed by studying the spatially modulated responses at steady-state. Of course, this gives us little insight about the dynamics in native cells but we believe that the regulations we found are meaningful. In other words, our work can be thought of as a step toward a comprehensive description of full spatiotemporal responses by going beyond bulk measurements (space is included). In order to present in greater details our distinction between spatial and temporal responses, we extended the discussion (page 21 lines 15-22).

5) On the same note, they discuss several times in the manuscript concepts about gradients involving diffusion of GTPases. To my knowledge, diffusion of many small GTPases including Rac and Cdc42 has been measured and usually has been found to be around $0.5 \mu\text{m}^2/\text{s}$ along the plasma membrane. Why can't they just use this value for their models or do they have reasons to believe that this value is not correct for Rac and CD42, or not correct in the leading edge? Is all their data compatible with this well-cited diffusion value? They need to add a paragraph discussing how their analysis helps us understand better how CDC42 and Rac diffuse in the membrane or simply use the literature values as a given and focus on the GEF and GAP gradients alone.

Our results are not in contradiction with the lateral diffusion coefficient reported for RhoGTPases. For Cdc42, the fact that we could not measure any difference between the activating optogenetic gradient and the PAK1 response gradient means that Cdc42 diffusion is not sufficient to smooth out the input gradient. Of course, our ability to discern two gradients depends on the resolution of our measurement. Taking two standard deviations of the measured spatial extent, we can estimate this resolution to be about $2 \mu\text{m}$. This means that the length scale due to Cdc42 diffusion cannot exceed $2 \mu\text{m}$. Using $0.5 \mu\text{m}^2/\text{s}$ for Cdc42 diffusion coefficient, this limit imposes an upper bound for the lifetime of Cdc42 in the GTP-bound state of 2s. This number may appear small, but: 1) this lifetime

has never been measured, and 2) it is recognized that RhoGTPases are actively cycling thanks to GAPs. Regarding our modeling approach, we wanted to stick to a minimal model and thus we neglected diffusion. Reaction-diffusion models are much more complicated and would not have led to any difference in the context of our study.

Thanks to the remark 5) of reviewer 2, we specified the quantitative implications of our results on Cdc42 given the diffusion coefficient reported in the literature (page 6 line 33-39).

6) Given the effects they see with β -chimaerin localization, which they interpret as retrograde flow of β -chimaerin, they should show whether this is a general effect and test whether other GEFs or GAPs in the front region are equally affected by such retrograde flow. What they may find would alter their model since it assumes that this is a specific transport mechanism for β -chimaerin. This is a main point in their paper and they need to show that this transport is specific for β -chimaerin and not a general phenomenon for all membrane localized proteins or other GEF and GAPs.

Actually, in the text we suggest that the retrograde flow is not acting on β -chimaerin itself, but rather on the molecular machinery that is required to limit its recruitment at the tip of the cell (page 9 lines 9-15). As discussed in the second paragraph of the discussion, such a machinery could be based on the advection by the retrograde flow of a factor responsible for the β -chimaerin membrane dissociation, or on coincident signaling between a factor promoting β -chimaerin membrane binding and a molecular maker of actin polymerization/branching. Concerning the specificity, as also said in the text, we propose that the mechanism that limits β -chimaerin recruitment to the tip is not specific only to β -chimaerin, since the Abi1 recruitment follows a similar pattern. Yet, we know that the retrograde flow is not acting on our synthetic opto-GEF constructs, since we have a very good match at the cell tip between the light patterns sent and the measured gradients that result from it (figure 2b).

7) More controls are needed at several places in the manuscript. For example, PAK1 is used as a reporter for Cdc42 and Rac1 activities. The authors should validate this approach by showing its distribution is not changed with optogenetic activation in the presence of drugs targeting Cdc42 or Rac1 (such as ZCL278 and NSC23766) and/or siRNA knockdown of Cdc42 and Rac1.

We agree with reviewer 2 that this control is important and it was missing in our first submission. We performed this control during the revision process, which is now presented in supplementary figure 2 and also mentioned in the main text (page 6 line 29-30).

8) The model and conclusions are drawn entirely from localizations of fluorescently tagged proteins (TIAM, ITSN, Abi1, PAK, β -chimaerin) under artificial polarization conditions. A quantitative confirmation using endogenous localizations of these proteins under naturally occurring polarization conditions is needed to ensure that these observed GEF/GAP distributions are relevant in cells during normal migration.

We refer here to our answer to point 2) of Reviewer 2.

9) The data shown in 3g needs some additional controls showing that the β -chimaerin enrichment is not a volume artifact from the lamellipodia (e.g. RFP alone). A similar control is needed for 3j to show that the lack of enrichment in protrusions isn't because there are no protrusions (alternatively an Arp2, or f-tractin staining could highlight protrusions). While the authors do not make a point about

temporal regulation, showing that areas with high β 2-chimaerin have instantaneous or delayed reduction in Rac1 signaling is needed to understand how its localization is regulated and how it regulates Rac.

To control that fluorescence enrichment is not a volume artifact due to the lamellipodia, we imaged cells expressing free iRFP, together with Lifeact-GFP in order to identify cell protrusions, as shown now in supplementary figure 7 and referenced in the main text (page 8 lines 39-41). Regarding the temporal regulation, as commented above in the point 4) of Reviewer 2, we did not consider it in the present work even if we agree that it would be very interesting to address it in future works.

10) Generally, the materials and methods are vague which makes interpretation of the data difficult. For example, despite describing a lengthy FRET normalization procedure, the authors report the FRET data in 3c and f as “non-normalized.” It is unclear then, how comparable the two curves really are. Similarly, in 6b, it appears the data has been somehow normalized such that ITSN at 4x is 1 but that is not stated in the methods. The authors should go over their manuscript and clarify how experiments were done and how data was processed and normalized in the different figures. This also would help one to understand how significant some of the claims are in the manuscript.

We agree that our description of curve normalization was not always clear in our manuscript. We fixed this point in the revised version (page 8 line 21-22, page 11 lines 7-9, lines 15-17, page 14 lines 3-4, page 14 line 11-12, page 24 lines 9-13, page 24 lines 17-28). Regarding FRET, we present non-normalized FRET data when the same FRET reporter was used in two different experimental conditions, and we present normalized FRET data when comparing two different reporters. We also refer here to our answer to Reviewer 3 point 5).

Concerning the figure 6b, the data are not normalized such that ITSN 4x is at 1. Our quantification of angular precision does not require normalization. The fact that the data point is very close to 1 and shows little variability means that cells are highly directional and very precise under the application of a sharp Cdc42 gradient. We added a small comment (page 16 line 27) to point reader's attention on this surprisingly low dispersed data point.

11) The localizations shown in 3b and 3d are much less pronounced than that of Pak1 in 2e, yet the quantification appears to show an even more pronounced decay; adding supplementary information that shows the cells in the figure quantified along the length of the cell with a color scale bar for reference. Alternatively, a single color-scale for 0-1 could be used for all images such that they are comparable across all figures.

The absolute levels of fluorescence of the data presented in 3b,d are lower than in 2d,f, which explains why the amplitude of the signal compared to the basal level appears lower in the figures, or equivalently why the basal level appears higher (the zero being set by the background outside cells). To clarify this point, we added colorbars next to the figures 2d,f and 3b,d and we explain our image normalization in the method section (page 24 line 27-30). As the normalized quantification shows in 3e, the localization of β 2-chimaerin is sharper than that of Pak1. This is possible since amplitude and decay length are two uncoupled features of gradients. We now also provide in supplementary figure 5 the quantification of the Abi1-iRFP images shown in 3b overlaid with the Pak1 quantification to show that, similarly to β 2-chimaerin, Abi1 recruitment is more localized toward the tip of the cell (commented in the main text page 8 lines 15-16).

Regarding the alternative proposition, we would like to point out that our images are normalized between 0 and 1 so the color-scale were already comparable in our first submitted manuscript.

12) Supplemental example images of several individual cells are needed showing how the gradients were applied and how the different steepnesses translated to the different measured gradients. How reproducible is this given the many feedbacks operating in cells? Also, a natural question is also how the gradients were actually realized in Pak1 and Cry2 recruitment. The authors need to show several image examples and gradient steepnesses analyzed in single cells and show how amplitude and steepness of illumination profiles changes the single cell recruitment profile.

The cell-to-cell reproducibility can be appreciated through the standard deviation given for each profile. Yet, for the sake of clarity, we also added a new **supplementary figure 3** to present the single cell data (mentioned **page 8 line 3** in the main text).

Reviewer #3 (Remarks to the Author):

In the current manuscript the authors use an optogenetic approach in combination with micro-patterning to characterize Cdc42 and Rac1 activation gradients in HeLa cells. Using optogenetic Rho GTPase activation of a confined part of the cell, they show that Rac1 and Cdc42 activity gradients show an exponential decay away from their area of activation. The Cdc42 activity patterns can be explained by the GEF activity patterns that the authors impose on the cell, combined with global GAP activity. However, following the observation that Rac1 activity peaks around 6 μ m from the cell edge after similar GEF stimulation, they propose a model where Rac1 is deactivated at the cell edge by a specific GAP, on top of global deactivating GAP activity. This specific Rac activation pattern is influenced by Cdc42 expression and actin dynamics based on siRNA and drug perturbations. Furthermore, the authors use a mobility assay to quantify cell polarization parameters and velocity following different optogenetic stimulation regimes with a Rac1 or Cdc42 GEF. The authors show that smaller Cdc42 activation patterns cause higher polarization precision, while the size of activity patterns for Rac1 does not seem to have an influence on velocity. They propose that the width of the activation pattern of the Rho GTPases is a crucial factor influencing velocity and polarization, as opposed to amplitude/slope? of the activity pattern.

The authors use an elegant approach to unravel very relevant open questions in the field. By confining the cell to a well-defined spatial pattern, and activating a GEF locally, questions of spatial Rho GTPase activity patterns / de-activating patterns can be elucidated. However, I feel that the manuscript lacks focus and the experimental descriptions are sometimes confusing. Also, while the experiments are technically sound, the interpretation of the data and the conclusions by the authors is not always justified. Another criticism is that while synthetic reconstitution of a signaling system captures some properties that resemble the native signaling system does not imply a causal relationship between the synthetic/native systems.

Please see points below.

We thank reviewer 3 for his/her comments and we hope that he/she will be satisfied by the improvements of the experimental descriptions, interpretations, and conclusions in the revised version of the manuscript. We answer point-by-point below.

Major points:

1) Did the authors check if the culturing of cells on the micropatterns had an influence on the Rho GTPase activity gradients mentioned in Figure 1 (Because cell shape, cytoskeleton and Rho GTPase activity all influence one another)? It would be a good control to show some figures of cells on micropatterns and, possibly, quantifications to prove that in these conditions Rac1 and Cdc42 polarity is lost.

This is a good point raised by Reviewer 3. It is very possible that the native Rac1 and Cdc42 gradients differ on micropatterns. Yet, our synthetic engineering approach shows that imposing GEF gradients let us recapitulate the quantitative features of the native Rac1 and Cdc42 gradients (as probed by the PAK1 gradients in figure 2e,g which are similar as the FRET gradient shown in figure 1b). Thus, the actual endogenous gradients on micropatterns are not of interest for us, since our aim is to understand the spatial properties of the native gradients (that we successfully mimic). It could be interesting, in the context of another work, to see whether Rac1 and Cdc42 gradients are actually similar on micropatterns or not. If it is not the case, it could imply that endogenous GEFs do not distribute as they do in a freely migrating cell, a fact that we circumvent with our synthetic approach.

We added a sentence on page 6 lines 43-45 to stress that the Rac1 induced gradient on micropatterns matched the native one in freely moving cells.

2) The authors setup a robust assay to unravel the role of Rac1 and Cdc42 gradients on cell migration. Through it, they demonstrated that Rac1 gradient is mostly involved in the control of cell speed, while Cdc42 mostly controls orientation precision. In addition to the reported data, the authors should also investigate, with the same approach, the role of the Cdc42/ β 2-Chimaerin/Rac1 crosstalk (fig. 2,3,4) on cell migration properties in order to demonstrate that the specific spatial profile of Rac1 activation is relevant for the control of cell speed and/or directionality (fig. 5,6). Therefore, I would suggest to perform the TIAM-CRY2 optogenetic-induced migration experiment on cells silenced for β 2-Chimaerin or Cdc42 (or, for the latter, its inhibition). The results of these experiments would further strengthen the main conclusions of the manuscript and provide a functional role for the above-mentioned crosstalks between the two GTPases.

We thank Reviewer 3 for this nice suggestion, and we performed the experiment. As shown in the new supplementary figure 9, when β 2-Chimaerin is suppressed and migration is induced by Rac1 (we used TIAM 4x condition) we see no effect on cell speed but a significant reduction in angular precision. This result suggests that the functional role of β 2-Chimaerin is to limit the Rac1 protrusive activity at the very cell front to allow Cdc42 activity to steer cell migration. We included this result in the revised manuscript (page 16 lines 37-42).

3) JLY drug cocktail experiment should be explained in more detail, how long before the measurement was the drug added, and for how long did this drug treatment last? Interference with the actin cytoskeleton will always drastically interfere with Rho GTPase signaling due to mechanical and biochemical feedback.

We now specify in the method section (page 23 lines 28-30) the details regarding our use of the JLY drug cocktail.

4) Do the authors have any idea which GAP or GAPs are involved in the “global GAP” activity?

We refer here to our answers to Reviewer 1 point 1) and Reviewer 2 point 2) that address this question (our short answer is no).

5) (Figure 4b) The modeling approach seems to be hampered by some incorrect assumptions, which leads to a model that could have been tweaked to match the experimental data: -e.g. The assumption that Rac and Cdc42 FRET amplitude measures can be directly compared in relation to GEF activity towards Rac1 and Cdc42 is wrong, as the FRET biosensors for different Rho GTPases contain different components with different affinities. Dynamic ranges of these sensors cannot be directly compared like this in terms of GEF activity.

-Why do the authors assume that the global GAP activity towards Cdc42 lower than for Rac1? (not measured or referred to)

-Only very indirect evidence is presented that Cdc42 has an influence on β 2-Chimaerin (not measured or referred to).

Reviewer 3 is right, and we thank him/her for raising a concern that we had missed. Indeed, FRET biosensors have different affinities and cannot be compared. As a consequence, we removed our previous supplementary figure 4 that showed non-normalized Rac1 and Cdc42 FRET profiles, since the absolute values are not comparable. Regarding our model, we removed our assumption regarding the relative GEF and GAP levels of Cdc42 versus Rac1. However, our modeling conclusions remain unchanged, since the activation and deactivation rates are arbitrary. Yet, we updated the figure 4 and the text (in red, pages 12-14) in the revised version of the manuscript to account for the modification in the values of rates.

Concerning the influence of Cdc42 on β 2-Chimaerin, we agree that it might be indirect, possibly going through our identified crosstalk between Cdc42 and Rac1 (as discussed in the manuscript). Yet, our experiments clearly show that Cdc42 acts on β 2-Chimaerin (figure 2e) and thus Rac1 (Figure 2c), and we think that these facts are sufficient for the objectives of the present work.

Figures:

1) Figure 1b, how are the lines for this quantification drawn in the cells, and are there multiple lines/cell? Figure 1a shows that a single cell contains many different gradients of activity along its membrane...

Indeed the gradients in Figure 1a present a significant variability within one cell and across different cells. However, in the context of our present work, we did not address this variability and we only focused on the global, averaged, features of these gradients. Thus, for our quantification we averaged two line scans per cell, where the lines were chosen in regions of the cell that show protrusive activity. We added a paragraph in the method section of the revised manuscript (page 24 lines 23-38) to explain how gradients were quantified.

2) Figure 2g (left) is not mentioned/explained in the legend. Which is the difference between upper and lower charts?

The two left plots in figure 2g correspond to two GEF gradients with different steepness (2x and 4x), as in figure 2e. We now precise it in the legend (page 8 line 1-3).

3) Fig 3:

-Legend of Fig. 3b mentions multiple experimental repeats, however figure shows a single example? Experimental result is not quantified.

The figure is the averaged image of the N=10 and 11 cells. We now explain in detail the averaging procedure in the new method section (page 24 line 23-38) and refer to it in the legend (page 11 line 13). Quantification is now given in supplementary figure 5.

-Legend of Fig 3d mentions "Micrographs represent the averaged fluorescence". What is averaged here?

As for the previous remark, we now explain in detail the averaging procedure in the new method section (page 24 lines 23-38) and refer to it in the legend (page 11 line 13).

-How is Fig. 3k quantified, no mention of ROI's, masks? The authors calculate statistics with a

Students T-test on the mean, while showing median values in the boxplot. Visualization of the population data should be accompanied with the appropriate statistical test (e.g. Wilcoxon or derivatives to compare median values). Same holds true for boxplots in fig 6.

We now provided details on the quantification of figure 3k in the legend (page 11 lines 28-30). We also changed our statistical test for the box plots presented in figures 3 and 6.

-Fig 3g: FRET measurements performed on fixed cells. Since Rho GTPase activity fluctuates on a timescale of seconds, a fixed sample measurement is not very convincing. Measuring and quantifying β 2-Chimaerin location (authors have the iRFP construct) and the Rac1 biosensor activity simultaneously in live-cell imaging would strengthen their conclusions.

Since our live imaging of FRET reporter are acquired in less than a second (200 ms), we do not expect a significant change between a FRET image from a fixed sample and a snapshot from a live movie. Yet, as a side note, the issue with the iRFP construct for live imaging of β 2-Chimaerin is that the reporter has a strong cytosolic and endomembrane staining. While it worked fine with optogenetic experiments since we could subtract the images before the activation, it did not show satisfying signals when we tried to use it as a live reporter of native processes.

4) Figure 6e-f: What is quantified on the y-axis?? The textual explanation of this figure is unclear. Do amplitude and slope mean the same thing? How is this measured / quantified / manipulated?

We now precise in the legend (page 19 line 20-22) the parameters being quantified (speed and angular precision, as in figure 6a and b) and we precise the difference between slope and amplitude. We also modified the Figures 6e-f to add a label on the y-axis.

Textual:

1) p2 Introduction: The whole concept of 'advection' should be described more clearly, and with correct references (introduction refers to Ref. 38-40, but there is no mention of advection of actin in those papers..). Authors should double check the references used, since some of them seem not to be connected to the preceding text, see below also.

We refer here to our answer to the minor point 3) of Reviewer 1.

2) p2 Introduction: What do the authors mean with "Rho GTPase futile cycles" on p2 introduction?

We removed the term "futile" that was not required here.

3) p2 Introduction: What do the authors mean by "such that the overall Rac1 gradient extent is increased" p2 introduction?

We now explain it better (page 3 line 33).

4) p1 Results: Which FRET biosensors are used in this study? If obtained from L. Hodgson, the authors incorrectly refer to Ref. 22 and 44 here.

The FRET biosensors used indeed come from L. Hodgson's lab, we corrected the citation.

5) p1 Results: "A canonical example is the synthesis-diffusion-degradation model (SDD).....the Bicoid morphogen gradient", the short mention of these models assumes much specialistic knowledge and should be detailed further if mentioned at all. (What is a "BMP gradient"??)

The SDD model is a standard model for describing the establishment of gradients. Since it is generic, it applies to many systems and at many scales. Here we refer to morphogen gradients because it has been extensively described in this context. Our work is interdisciplinary, and we like the principle of connecting different fields that involve the same concepts.

Regarding the BMP gradient, we have detailed the acronym (page 4 line 29).

6) Discussion p1: Authors write: "...we reveal the topology of the interaction network between Cdc42, Rac1 and their regulators...etc" This statement is too general, 2 well-characterized GEFs (out of multiple GEFs for Rac1/Cdc42 involved in cell migration) were manipulated and investigated. Authors should temper their claims.

We have tempered our claim, indeed we propose one network topology and showed that it is sufficient to recapitulate the observed gradients (page 20 lines 3-5).

7) Discussion p1: The authors state: "The spatial properties of these gradients are similar to what has been observed before in other cell types" and refer to Ref 15,45. Ref 15. refers to a paper using the Raichu probe, which is constitutively located at the plasma membrane, and thus cannot be used to measure spatial gradients into the cell. Ref. 45 refers to a paper where it is shown that Rac1 has the highest activity at the cell border and during lamellopodia formation. References seem not to support the statement by the authors.

Rho GTPases are activated at the plasma membrane, such that their gradients are along the membrane. In Ref 15, indeed the probe is constitutively anchored at the plasma membrane, but the only difference with probes developed later is the sensitivity to GDI that extract Rho GTPases from the membrane. We do not see why Raichu probes "cannot be used to measure spatial gradients into the cell" since it is actually the main topic of Ref 15. Regarding Ref 45, their figure 4F clearly shows a Rac1 activity gradient that peaks $\sim 4\mu\text{m}$ away from the cell edge. The gradient presents a very similar shape as the one we quantified here.

9) Discussion p2: The authors state: "...the response functions we measured under our steady optogenetic activations did not show evolving spatiotemporal patterns (supp fig 5)" It is unclear what the authors mean by this statement, and how it is supported by supp. fig. 5?

We mean that the spatial properties of the response are not changing as a function of time.

Minor points:

1) Write in the "FRET" part of "Material and Methods" section the description of Cdc42-FRET-biosensor experiments and where it was obtained.

Indeed we had forgotten this biosensor in the FRET part of Materials and Methods. The procedure and provenance of this biosensor is the same as for the Rac1-FRET biosensor, the text was corrected to mention it (page 24 line 2).

2) Working mechanism of BCN-RGD assay should be shortly explained in Results text at figure 5.

We now explain shortly the mechanism (page 14 lines 22-25).

3) Assay in figure 5 would more appropriately be called a "spreading" assay, since no moving cells are followed over time as in a migration assay.

For the sharp Rac1 and Cdc42 gradients, the back of the cell is also moving toward the imposed direction such that the whole cell body moves, as well as the nucleus. As explicitly said in the text (page 14 lines 35-38), we observe in our assay a transition from oriented spreading to migration, which justifies for us the appellation of a migration assay. In addition, in the discussion (page 21 lines 26-43) we compare our measurement of migration parameters to other measurements defined on longer time scales, where cells move more substantially.

In its current state, I would not recommend a publication of this manuscript in NCB. Overall the manuscript should be improved in terms of clarity of description of the experimental procedures and use of jargon, to make it more accessible to a broader audience. In its present state, it seems to be aimed at a very specific reader niche. The manuscript should also strengthen its conclusions and focus by adding experiments and clarifications mentioned under “Major points” and “Figures”.

If improved, I think it would be very suitable for publication in a more technical journal, e.g. Biophysical Journal.

We hope that the revised version of manuscript will now satisfy Reviewer 3. Regarding the suggestion of publishing the work in a technical journal, we disagree. As said by Reviewer 3 (“The authors use an elegant approach to unravel very relevant open questions in the field”) and also recognized by Reviewers 1 and 2, we believe that both our biological findings and conceptual approach justify a publication in a large audience journal. Indeed, the notion of spatial gradients in cell biology is fundamental. At the multi-cellular level, spatial gradients ensure the coordination of cellular activities, while at the cellular level they ensure the spatiotemporal orchestration of multiple biochemical activities, giving rise to a cellular-scale order. Our work constitutes a first example in which quantitative optogenetics are used to spatially dissect the functional architecture of an intracellular network.

REVIEWERS' COMMENTS:

Reviewer #3 (Remarks to the Author):

The manuscript has been improved significantly in terms of data presentation and clarity, I thank the authors for addressing the concerns raised by me and the other reviewers. The approach used in this study is of interest for the cell biology community and the paper in this form advances the field.

However, I would recommend the authors to double check the whole manuscript again for spelling and precision.

e.g:

-P1L21 "micopatterning" should be changed to "micropatterning"...

-In the description of figure 6 in the "Results" section of the manuscript figure 6f,g,h are still called, while the new figure 6 only shows a-e...

an optional suggestion: Adding an overlay of the f-actin and b-chimaerin channels (highlighting colocalization) in the new supp. figure 7 would increase interpretation for the reader.